# Linker histone H1 prevents R-loop accumulation and genome instability in heterochromatin

Aleix Bayona-Feliu[1,2,3], Anna Casas-Lamesa[1,2], Oscar Reina[2], Jordi Bernués[1,2] & Fernando Azorín[1,2]

Linker histone H1 is an important structural component of chromatin that stabilizes the nucleosome and compacts the nucleofilament into higher-order structures. The biology of histone H1 remains, however, poorly understood. Here we show that *Drosophila* histone H1 (dH1) prevents genome instability as indicated by the increased γH2Av (H2AvS137P) content and the high incidence of DNA breaks and sister-chromatid exchanges observed in dH1-depleted cells. Increased γH2Av occurs preferentially at heterochromatic elements, which are upregulated upon dH1 depletion, and is due to the abnormal accumulation of DNA:RNA hybrids (R-loops). R-loops accumulation is readily detectable in G1-phase, whereas γH2Av increases mainly during DNA replication. These defects induce JNK-mediated apoptosis and are specific of dH1 depletion since they are not observed when heterochromatin silencing is relieved by HP1a depletion. Altogether, our results suggest that histone H1 prevents R-loops-induced DNA damage in heterochromatin and unveil its essential contribution to maintenance of genome stability.

[1] Institute of Molecular Biology of Barcelona, IBMB, CSIC, Baldiri Reixac, 4, 08028 Barcelona, Spain. [2] Institute for Research in Biomedicine, IRB Barcelona, The Barcelona Institute of Science and Technology, Baldiri Reixac, 10, 08028 Barcelona, Spain. [3]Present address: Centro Andaluz de Biología Molecular y Medicina Regenerativa-CABIMER, Universidad de Sevilla-CSIC-Universidad Pablo de Olavide, 41092 Seville, Spain. Aleix Bayona-Feliu and Anna Casas-Lamesa contributed equally to this work. Correspondence and requests for materials should be addressed to J.B. (email: jordi.bernues@ibmb.csic.es) or to F.A. (email: fambmc@ibmb.csic.es)

Histones are chromosomal proteins that play an important structural function in packaging of the eukaryotic genome into chromatin. Histones H2A, H2B, H3 and H4 form an octameric complex that organizes 146 bp of DNA and constitutes the protein core of the nucleosome, a highly conserved particle that is the basic structural and functional subunit of chromatin. In addition, linker histones H1 bind to the nucleosome core particle and stabilize folding of the nucleofilament into higher-order structures[1, 2]. Histones are also crucial to the regulation of genomic functions. In recent years, we have become aware of the essential contribution of core histones to the epigenetic regulation of multiple genomic processes from RNA transcription to DNA replication, recombination and repair, chromosome segregation and genome integrity. In comparison, the biology of linker histones H1 remains poorly understood. Histones H1 are less well conserved than core histones and, in metazoa, they generally exist in multiple variants that play partially redundant functions. For instance, deletion of one or two of the seven mouse somatic H1 variants have no detectable effects

since mice develop normally and show normal total H1 levels due to the compensatory expression of other variants[3, 4]. Mice also contain one female and three male germline specific H1 variants[5]. Humans show a similar complexity. In contrast, H1 diversity in *Drosophila melanogaster* is low since it contains only one somatic (dH1) and a second germline specific (dBigH1) variant[5, 6]. Note, however, that somatic dH1 is encoded by a multigene family, which is unusual since, in most metazoa, H1 variants are encoded by single-copy genes[6]. Unicellular eukaryotes such as *Saccharomyces cerevisiae*, *Aspergillus nidulans* and *Tetrahymena termophila* also contain a single histone H1-like gene[7–9].

Genetic studies unveiled the essential contribution of histones H1 to metazoan development. In *Drosophila*, dH1 depletion impairs normal development and causes lethality[10, 11]. Similarly, triple H1 variants knockout mice fail to compensate total H1 levels and die at early developmental stages[12]. The reason for this lethality is not well understood. On one hand, although they are widely distributed across the genome, histones H1 have a weak contribution to gene expression since depletion of *Drosophila*

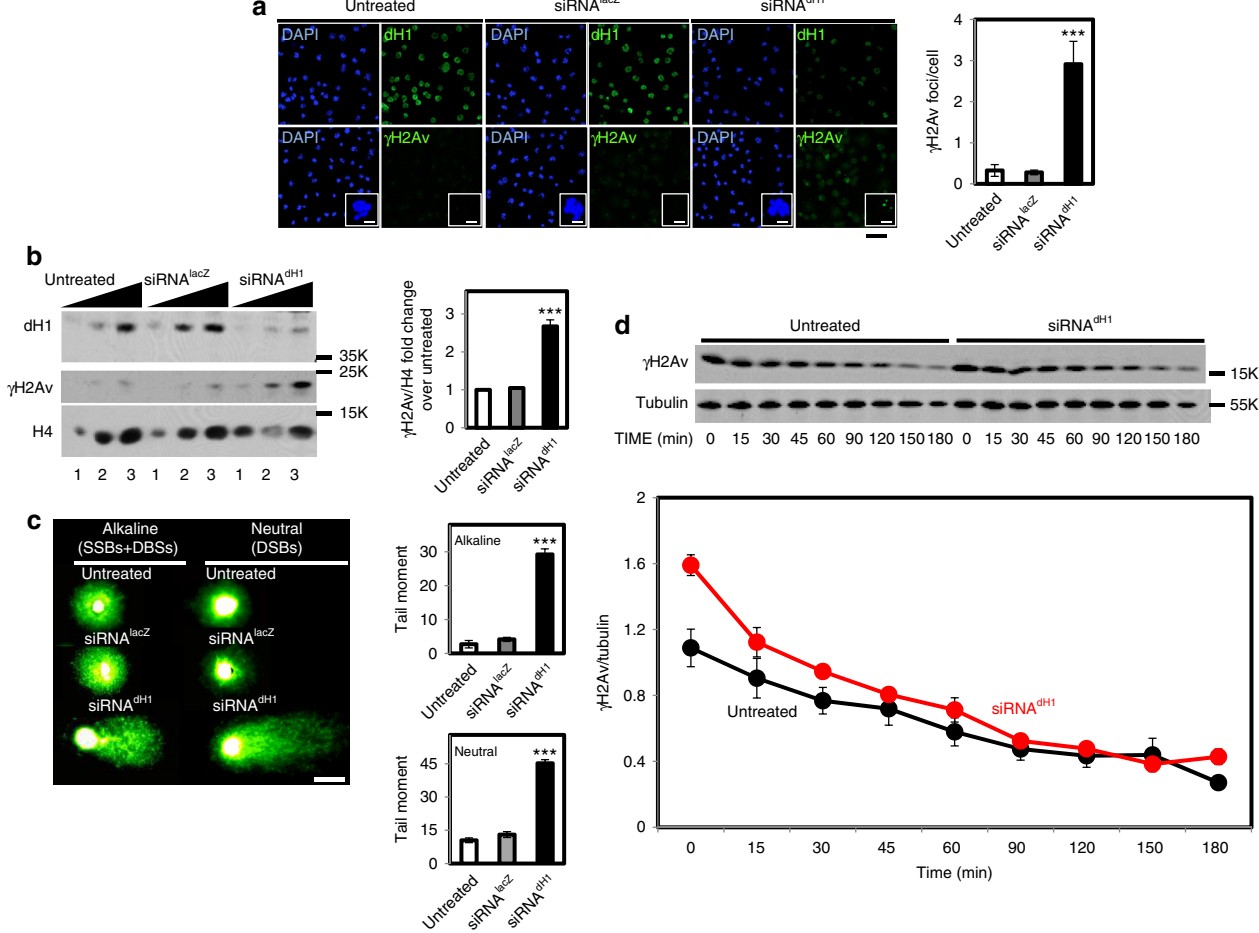

**Fig. 1** dH1 depletion induces DNA damage. **a** Immunostaining of dH1-depleted (siRNA$^{dH1}$) and control undepleted cells (siRNA$^{lacZ}$ and untreated) with αdH1 and αγH2Av antibodies (both in *green*). DNA was stained with DAPI (*blue*). *Insets* show enlarged images of representative individual cells. *Scale bars* are 20 μm and 2 μm in the *Insets*. On the *right*, the number of γH2Av foci per cell is presented (n > 100 for each condition). *Error bars* are s.e.m. The p-value of siRNA$^{dH1}$ respect to siRNA$^{lacZ}$ is indicated (***p<0.005; two-tailed Student's t-test). **b** WB analyses with αdH1, αγH2Av and αH4 of increasing amounts of extracts (lanes 1–3) prepared from siRNA$^{dH1}$, siRNA$^{lacZ}$ and untreated cells. The positions corresponding to molecular weight markers are indicated. On the *right*, quantitative analysis of the results (N = 3). *Error bars* are s.e.m. The p-value of siRNA$^{dH1}$ respect to siRNA$^{lacZ}$ is indicated (***<0.005; two-tailed Student's t-test). **c** Alkaline and neutral single-cell electrophoresis analyses of siRNA$^{dH1}$, siRNA$^{lacZ}$ and untreated cells. *Scale bar* corresponds to 20 μm. On the *right*, relative comet-tail moments are presented (n > 100 for each condition). *Error bars* are s.e.m. The p-values of siRNA$^{dH1}$ respect to siRNA$^{lacZ}$ are indicated (***<0.005; two-tailed Student's t-test). **d** On the *top*, WB analysis with αγH2Av and αtubulin at different time points after X-ray irradiation (10 Gy) of siRNA$^{dH1}$ and untreated cells. The positions corresponding to molecular weight markers are indicated. On the *bottom*, quantitative analysis of the results (N = 3)

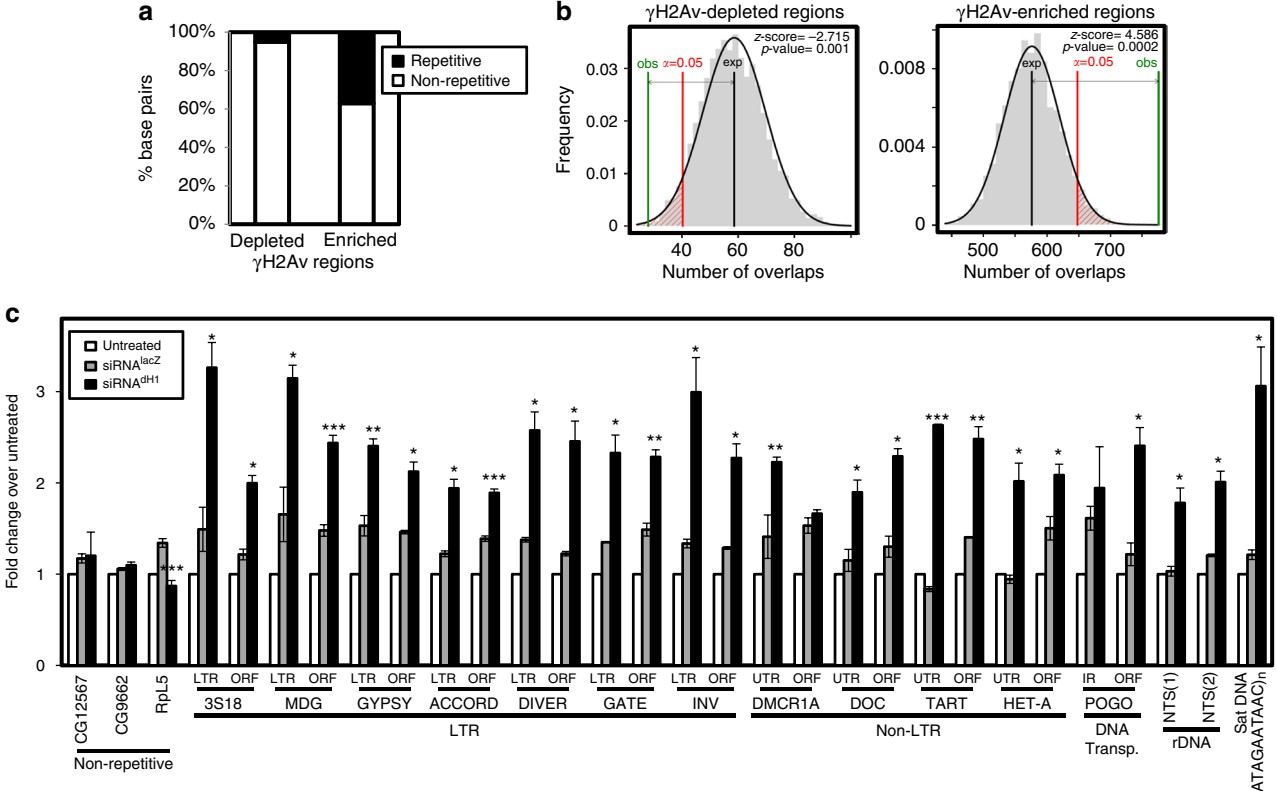

**Fig. 2** DNA damage induced by dH1 depletion occurs preferentially in heterochromatin. **a** The proportion of base pairs (bp) matching to repetitive and non-repetitive elements, as determined by RepeatMasker analysis, are presented for regions showing specific γH2Av enrichment and depletion in dH1-depleted cells respect to control untreated cells. **b** Permutation experiments showing statistical significance of the enrichment in repeated DNA sequences of the regions showing specific γH2Av enrichment (*right*) and depletion (*left*) in dH1-depleted cells respect to control untreated cells. The frequency of the number of overlaps with repetitive DNA elements, as determined by the regioneR package using the UCSC dm3 RepeatMasker track (March 2017), is presented based on 5000 random permutations of the experimentally identified regions. The average expected number of overlaps (*black*) is compared with the observed number of overlaps (*green*). The $\alpha = 0.05$ confidence interval is indicated (*red*). z-scores and permutation test p-values of the differences are also indicated. **c** γH2Av ChIP-qPCR analyses at the indicated repetitive and non-repetitive regions in siRNA[dH1], siRNA[lacZ] and untreated cells (N = 2). For each position, the fold-change respect to the untreated condition at this position is presented. *Error bars* are s.e.m. The p-values of siRNA[dH1] respect to siRNA[lacZ] are indicated (no asterisk >0.05, *<0.05, **<0.01, ***<0.005; two-tailed Student's t-test)

dH1 affects expression of <10% of protein-coding genes[11]. Similarly, only a few hundreds of genes are found deregulated in ES-cells derived from triple H1 knockout mice[13], whose total H1 content is reduced by ∼50%. Furthermore, in *S. cerevisiae*, deletion of the single H1-like gene *Hho1* affects expression of only a few genes[14]. In contrast to its weak effect on expression of euchromatic genes, dH1 depletion strongly affects silencing of transposable elements (TE) and other repetitive DNA sequences[11, 15], and induces DNA damage[11]. However, the actual molecular mechanisms underlying these defects remain unknown.

Here, we show that dH1 depletion induces the accumulation of R-loops in heterochromatin. R-loops are three-stranded structures formed when a newly synthesized RNA forms a DNA:RNA hybrid with the transcribed strand leaving the untranscribed strand single-stranded (ssDNA). R-loops can form naturally during plasmid and mitochondrial DNA replication[16, 17], in immunoglobulin class switching[18] or during transcription[19]. However, unregulated R-loops formation is also an important source of genomic instability[20–23]. In particular, R-loops can stall replication fork progression, which generates double-stranded breaks (DSBs) and induces hyperrecombination[24, 25]. Our results show that DNA damage, genomic instability and apoptosis induced by dH1 depletion depend on R-loops. We also show that dH1 contribution to R-loops

dynamics is specific since HP1a depletion, which strongly relieves heterochromatin silencing too[26], does not induce R-loops accumulation.

## Results

**dH1 depletion induces DNA damage in heterochromatin.** dH1 depletion in the *Drosophila* wing imaginal disc was shown to induce DNA damage, as determined by an increased αγH2Av reactivity[11]. However, the underlying molecular mechanisms of this effect remained unknown. Here, to address this question, we performed depletion experiments in cultured *Drosophila* S2 cells, which constitute a homogeneous cell population amenable to mechanistic studies. As previously observed in the wing imaginal disc, RNAi-induced dH1 depletion in S2 cells significantly increased γH2Av content, as determined by both immunofluorescence (IF) (Fig. 1a) and western blot (WB) analyses (Fig. 1b). This increase was not observed in control cells treated with siRNAs against LacZ (Fig. 1a and b), and it was not associated with increased total H2Av content (Supplementary Fig. 1). Concomitant to increased γH2Av, dH1-depleted cells showed a high incidence of DNA breaks (DBs), as determined by the increased tail-moment observed in single-cell electrophoretic analyses performed under alkaline and neutral conditions to detect both single-stranded (SSBs) and double-stranded DNA breaks (DSBs) or only DSBs, respectively (Fig. 1c). In addition,

**Table 1 Summary of repetitive elements showing specific γH2Av and R-loops enrichment in dH1-depleted cells**

| Retrotransposons | | | | |
|---|---|---|---|---|
| LTR | | Non-LTR | DNA transposons | simple repeated satellite DNA sequences |
| ACCORD | HMSBEAGLE | BS[a] | BARI[a] | $(AAGAG)_n$/$(AAGAGAG)_n$ and related repeats |
| BICA[a] | IDEFIX | DMCR1A | DNAREP | $(AACAC)_n$ and related repeats[b] |
| BLOOD[a] | INVADER | DMRT1[a] | FB[a] | $(AATAT)_n$ and related repeats |
| BURDOCK[a] | MAX[a] | DOC | HOBO[a] | $(AATAGAC)_n$[b] |
| CHIMPO[b] | MDG | FW | LOOPER[a] | $(AATAACATG)_n$/Prod[b] |
| CHOUTO[a] | MICROPIA | HET-A | MARINER[a] | $(ATAGAATAAC)_n$ |
| CIRCE[a] | NOMAD[a] | IVK[a] | M4DM[a] | $(CAACTTT)_n$[a] |
| COPIA | QUASIMODO[a] | JOCKEY[a] | PROTOP | $(TAGA)_n$[b] |
| DIVER | ROO | LINEJ[a] | S[a] | $(TCGGA)_n$[a] |
| DM1731[a] | STALKER | R1 | TC[a] | $(TGC)_n$[a] |
| DM176 | TABOR[b] | TART | TRANSIB[a] | rDNA NTS[c] |
| DM297 | TLD[a] | | POGO | |
| FROGGER[a] | TOM[a] | | | |
| GATE | TRANSPAC[a] | | | |
| GTWIN[a] | ZAM | | | |
| GYPSY | 3S18 | | | |

[a]Only γH2Av enrichment detected
[b]Only R-loops enrichment detected
[c]Tandem repeats of the non-transcribed rDNA spacer

radiation-induced DNA damage was repaired efficiently in dH1-depleted cells (Fig. 1d), indicating that DNA repair was not impaired. Triple H1 knockout mouse ES-cells also showed efficient DNA repair[27].

Next, we performed γH2Av ChIP-seq analyses to determine the genomic regions where dH1 depletion induced DNA damage. In these experiments, we detected a similar number of γH2Av-enriched regions in control and dH1-depleted cells, a large proportion of which mapped to common targets (Supplementary Fig. 2a). γH2Av preferentially localized to promoters (Supplementary Fig. 2b and c) of actively transcribed genes (Supplementary Fig. 2d), suggesting a role of γH2Av in transcription regulation. In this regard, Jil1 has been shown to phosphorylate H2Av at promoters at the same residue S137 (γH2Av) that becomes phosphorylated in response to DNA damage[28]. Jil1-dependent H2Av phosphorylation stimulates PARP-1 activity and promotes transcription[28]. Notice that H2Av is the *Drosophila* homolog of both mammalian H2A.X and H2A.Z[29, 30]. Differential enrichment analysis detected 166 regions where γH2Av content was significantly increased in dH1-depleted cells, in front of 26 regions where γH2Av content was reduced upon dH1 depletion. Regions showing specific γH2Av enrichment in dH1-depleted cells had a high content on repetitive DNA elements, including TE and simple repeated satellite DNA sequences (Fig. 2a). As a matter of fact, ~ 70% of these regions contained repeated DNA sequences. This enrichment was statistically significant, as shown by simulation experiments where the expected overlap with repetitive elements, based on 5000 genome-wide random permutations of the identified regions, was compared to the actual overlap observed (permutation test *p*-value <0.0002; Fig. 2b). ChIP-qPCR experiments confirmed these results since dH1 depletion increased γH2Av content at multiple repetitive elements, including various LTR and non-LTR retrotransposons, DNA transposons, a satellite DNA and two repetitive regions of the non-transcribed rDNA spacer (NTS), whereas no increase was observed at several non-repetitive regions analyzed (Fig. 2c). In contrast, regions where γH2Av was differentially reduced in dH1-depleted cells were poor in repeated DNA sequences (Fig. 2a) and preferentially mapped to promoters of active genes (Supplementary Fig. 3a and b), likely reflecting the downregulation that dH1 depletion causes

on expression of a reduced set of genes[11]. Table 1 summarizes the repetitive DNA elements at which differential γH2Av enrichment was specifically detected in dH1-depleted cells.

Repetitive DNA elements preferentially localize at heterochromatic regions. In particular, most TE copies and satellite DNAs locate at centromeric heterochromatin[31, 32]. HET-A and TART retrotransposons also localize to telomeric regions[33]. Consequently, results reported above suggest that DNA damage induced by dH1 depletion preferentially accumulates in heterochromatin. IF experiments were consistent with these results since the overlap of γH2Av foci with HP1a, which marks heterochromatin, was significantly higher in dH1-depleted than in control cells (Supplementary Fig. 4a). Notice that the highest γH2Av/HP1a overlap increase was observed in cells containing only one γH2Av foci, indicating that it did not reflect random colocalization due to the increased number of γH2Av foci detected in dH1-depleted cells. Notably, while αγH2Av reactivity was unusual in metaphase chromosomes from control cells, it was significantly more frequent in dH1-depleted metaphase chromosomes (Supplementary Fig. 4b). In some cases, very intense αγH2Av signals were observed that tend to localize at pericentromeric regions, excluding the actual centromeres, and at telomeric regions (Supplementary Fig. 4c).

Concomitant with intense DNA damage, we detected an increased frequency of sister-chromatid exchanges (SCE) in dH1-depleted cells (Fig. 3a). In addition, dH1-depleted cells showed a high incidence of chromosome segregation defects (Fig. 3b) and, in particular, of anaphase chromatin bridges.

**dH1 depletion induces R-loops accumulation in heterochromatin.** Heterochromatic elements, which are normally silenced, become strongly upregulated upon dH1 depletion[11, 15]. In this context, we hypothesized that the unregulated expression of heterochromatic transcripts in dH1-depleted cells might cause their abnormal retention in chromatin and, thus, facilitate R-loops formation[20–23]. In this regard, IF experiments with S9.6 antibodies, which recognize R-loops[34], showed significantly increased S9.6 reactivity in dH1-depleted cells in comparison to control untreated cells or treated with siRNAs against LacZ (Fig. 4a and b, top). S9.6 immunostaining was strongly reduced by co-expression of human RNase H1 (RNH1), which targets

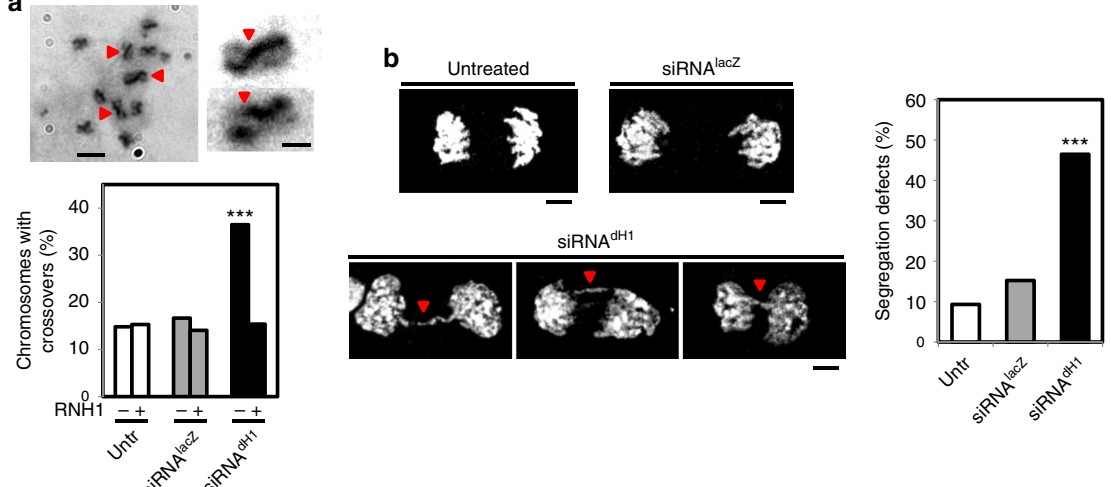

**Fig. 3** dH1-depleted cells show a high frequency of SCE and chromosome segregation defects. **a** SCE-assay in siRNA[dH1] cells. Sister chromatids are identified on the basis of their differential Giemsa staining due to their twofold difference in BrdU incorporation. Enlarged images of representative examples are shown on the *right. Scale bars* are 6 μm and 3 μm in the enlarged images. *Red arrowheads* indicate crossovers. On the *bottom*, the percentages of chromosomes with crossovers are presented for siRNA[dH1], siRNA[lacZ] and untreated cells expressing human RNH1 (+) or not (−) (n > 100 for each condition). The *p*-values of siRNA[dH1] respect to siRNA[lacZ] are indicated (no asterisk >0.05, ***<0.005; two-tailed Fisher's exact *F*-test).
**b** Anaphase figures for siRNA[dH1], siRNA[lacZ] and untreated cells stained with DAPI (in *white*) are presented. *Scale bars* are 5 μm. *Red arrowheads* indicate chromatin bridges. On the *right*, the percentages of mitoses showing segregation defects are presented for siRNA[dH1], siRNA[lacZ] and untreated cells (n > 40 for each condition). The *p*-value of siRNA[dH1] respect to siRNA[lacZ] is indicated (***<0.005; two-tailed Fisher's exact *F*-test)

DNA:RNA hybrids[35] (Fig. 4a and b, top), as well as by treatment with cordycepin, which inhibits RNA synthesis and, thus, abolishes R-loops formation[25, 36] (Supplementary Fig. 5). Noteworthy, αγH2Av reactivity of dH1-depleted cells was highly reduced by both RNH1 expression (Fig. 4a and b, center) and cordycepin treatment (Supplementary Fig. 5), showing significant overlapping with S9.6 reactivity (Fig. 4b, bottom). Furthermore, the high frequency of SCE detected in dH1-depleted cells was strongly reduced upon RNH1 expression (Fig. 3a). Altogether these results suggest that DNA damage and genomic instability induced by dH1 depletion are associated with R-loops accumulation.

R-loops-induced DNA damage is usually associated with DNA replication[24, 25]. Therefore, we analyzed whether αγH2Av reactivity observed in dH1-depleted cells was linked to DNA replication. For this purpose, we performed IF experiments in G1-, S- and G2/M-phase sorted cells. In dH1-depleted cells, γH2Av was low at G1-phase, strongly increased at S-phase to decrease again at G2/M-phase (Fig. 4c, top). Notice that, in all phases, αγH2Av reactivity of dH1-depleted cells was higher than in control undepleted cells, being highly reduced upon RNH1 expression (Fig. 4c, top). Note also that γH2Av persisted at G2/M-phase (Fig. 4c, top), which is in agreement with the increased αγH2Av reactivity observed in metaphase chromosomes of dH1-depleted cells (Supplementary Fig. 4b and c). WB analyses confirmed these results since, with respect to control cells, γH2Av content of dH1-depleted cells increased more in S-phase than in G1- or G2/M-phase (Fig. 4d). Notably, dH1-depleted cells showed high S9.6 reactivity at G1-phase that was strongly reduced by RNH1 expression (Fig. 4c, bottom), suggesting that R-loops induced by dH1 depletion were abundant at G1-phase. S9.6 reactivity was also intense in S-phase in both control and dH1-depleted cells. However, although S9.6 reactivity of S-phase sorted cells was significantly reduced upon treatment with bacterial RNH in vitro (Supplementary Fig. 6), it was largely resistant to RNH1 expression in vivo (Fig. 4c, bottom and

Supplementary Fig. 6) and, furthermore, treatment with RNase A abolished S9.6 reactivity of S-phase cells that persisted after RNH treatment in vitro (Supplementary Fig. 6; Discussion section).

Next, we performed genome-wide DNA:RNA hybrids immunoprecipitation experiments (DRIP-seq) to determine the genomic regions where R-loops accumulate upon dH1 depletion. For this purpose, genomic DNAs prepared from control untreated and dH1-depleted cells were subjected to immunoprecipitation with S9.6 antibodies before and after treatment with bacterial RNase H (RNH) to remove R-loops. Immunoprecipitation with S9.6 antibodies detected a high number of enriched genomic regions in both control and dH1-depleted cells, which strongly overlapped (Supplementary Fig. 7a). These regions, which distributed along the entire genome (Supplementary Fig. 7b) and located at both coding and intergenic regions (Supplementary Fig. 7c), were strongly reduced upon RNH treatment (Supplementary Fig. 7c), suggesting that a significant proportion of the genome is involved in R-loops formation. Others obtained similar results in mammalian cells[37, 38]. Differential enrichment analysis of the regions showing >30% reduction on coverage after RNH treatment detected 189 regions where R-loops abundance specifically increased in dH1-depleted cells, in front of only 14 where R-loops abundance was reduced upon dH1 depletion. R-loops regions specific of dH1-depleted cells were highly enriched in repetitive DNA sequences (Fig. 5a), both TE and satellite DNAs, which were detected in ~ 95% of them. Permutation experiments corroborated the strong enrichment in repetitive DNA elements of the R-loops regions specific of dH1-depleted cells (permutation test *p*-value <0.0002; Fig. 5b). DRIP-qPCR experiments confirmed these results since dH1 depletion significantly increased the proportion of repetitive DNA sequences immunoprecipitated by S9.6 antibodies and RNH treatment abolished this increase (Fig. 5c). Table 1 summarizes the repetitive DNA elements at which differential R-loops enrichment was specifically detected in dH1-depleted cells. These results suggest that, similar to γH2Av, R-loops

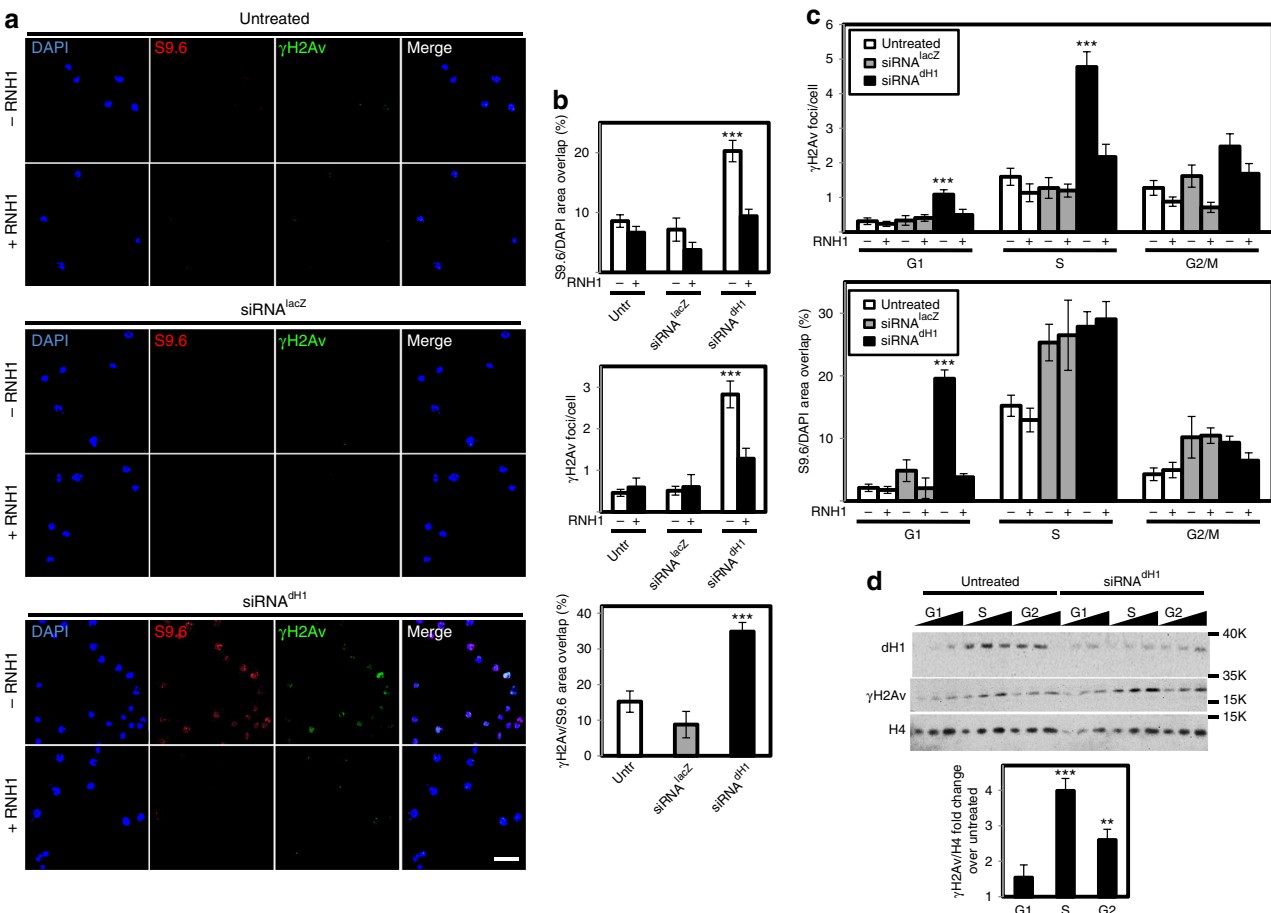

**Fig. 4** DNA damage induced by dH1 depletion associates with R-loops accumulation. **a** Immunostainings with αγH2Av (*green*) and S9.6 (*red*) antibodies of siRNA^dH1, siRNA^lacZ and untreated cells overexpressing human RNH1 (+) or not (−). *Scale bar* corresponds to 10 μm. **b** Quantitative analysis of the results shown in **a**. S9.6 (*top*) and γH2Av (*center*) reactivities determined as the proportion of DAPI area stained with S9.6 antibodies and the number of γH2Av foci per cell are presented (n > 50 for each condition). On the *bottom*, the extent of γH2Av/S9.6 colocalization is presented as the proportion of γH2Av area overlapping with S9.6 reactivity (n > 50 for each condition). *Error bars* are s.e.m. The *p*-values of siRNA^dH1 respect to siRNA^lacZ are indicated (no asterisk >0.05, ***<0.005; two-tailed Student's *t*-test). **c** αγH2Av (*top*) and S9.6 (*bottom*) reactivities of G1-, S- and G2/M-phase sorted siRNA^dH1, siRNA^lacZ and untreated cells overexpressing RNH1 (+) or not (−) are presented as in **b** (n > 50 for each condition). *Error bars* are s.e.m. The *p*-values of siRNA^dH1 respect to siRNA^lacZ are indicated (no asterisk >0.05, ***<0.005; two-tailed Student's *t*-test). **d** WB analyses with αdH1, αγH2Av and αH4 antibodies of siRNA^dH1 and untreated cells sorted at G1-, S and G2/M-phase. The positions corresponding to molecular weight markers are indicated. Quantitative analysis is shown on the *bottom* (N = 2). *Error bars* are s.e.m. The *p*-values respect to untreated are indicated (no asterisk >0.05, **<0.01, ***<0.005; two-tailed Student's *t*-test)

accumulate in heterochromatin upon dH1 depletion. IF experiments performed in polytene chromosomes from dH1-depleted *his1*^RNAi flies were consistent with these results since we detected high S9.6 reactivity in the heterochromatic chromocenter that was abolished by RNH1 expression (Supplementary Fig. 8).

We also performed labeling experiments to analyze the effects of dH1 depletion on DNA replication. Double IdU/CldU pulse labeling showed that, compared to control cells, DNA replication progressed slowly in dH1-depleted cells, as determined from the reduced length of the 2nd label (Fig. 6a, left). In addition, the ratio of the lengths of the two labels (asymmetry index), which measures the propensity of replication to stall[24, 39], increased (Fig. 6a, right). RNH1 expression abolished these defects, suggesting that they are associated with R-loops formation. We also observed that dH1 depletion induced significant EdU incorporation in heterochromatin during early S-phase, while it was constrained to mid/late S-phase in control cells (Fig. 6b and c), suggesting that heterochromatin replicates earlier in dH1-depleted cells than in control cells.

**HP1a depletion does not induce R-loops accumulation.** Next, we addressed whether the accumulation of R-loops detected in heterochromatin of dH1-depleted cells was simply the consequence of the relief of silencing caused by dH1 depletion[10, 11, 15]. For this purpose, we performed depletion of HP1a, an essential heterochromatin component that is also required for silencing[26]. Despite the strong upregulation of heterochromatin elements observed in HP1a-depleted cells (Supplementary Fig. 9a), no significant S9.6 reactivity was detected by IF (Fig. 7a). Similarly, S9.6 reactivity at the heterochromatic chromocenter of HP1a-depleted polytene chromosomes remained low in comparison to that observed in dH1-depleted polytene chromosomes (Supplementary Fig. 9b). DRIP-qPCR experiments confirmed these results since no significant R-loops were detected at heterochromatic elements in HP1a-depleted cells (Fig. 7c). Concomitant to the lack of S9.6 reactivity, HP1a depletion did not increase γH2Av content, as determined by IF (Fig. 7a) and WB analyses (Fig. 7b). Furthermore, HP1a depletion in the wing imaginal disc did not induce the strong apoptotic wing phenotype observed upon dH1 depletion[11] (Supplementary Fig. 9c), which is

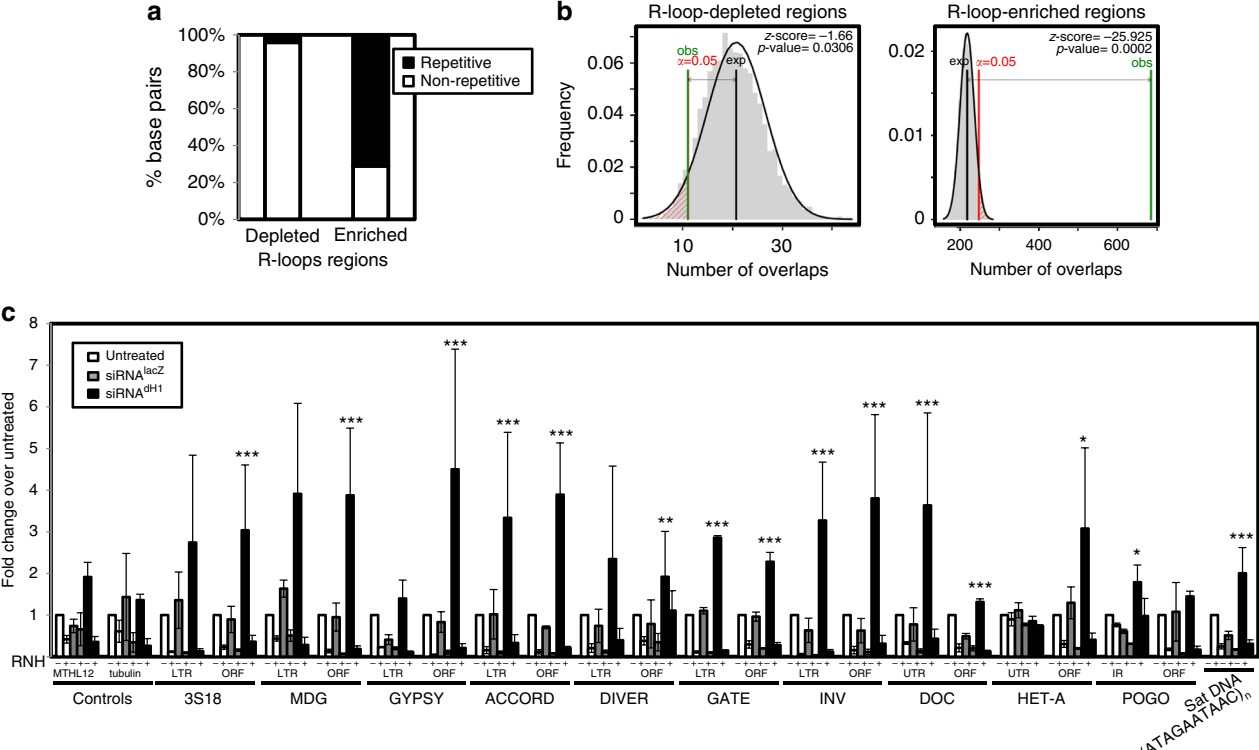

**Fig. 5** R-loops induced by dH1 depletion accumulate in heterochromatin. **a** The proportion of base pairs (bp) matching to repetitive and non-repetitive elements, as determined by RepeatMasker analysis, are presented for regions showing specific R-loops enrichment and depletion in dH1-depleted cells respect to control untreated cells. **b** Permutation experiments showing statistical significance of the enrichment in repeated DNA sequences of the regions showing specific R-loops enrichment (*right*) and depletion (*left*) in dH1-depleted cells respect to control untreated cells. The frequency of the number of overlaps with repetitive DNA elements, as determined by the regioneR package using the UCSC dm3 RepeatMasker track (March 2017), is presented based on 5000 random permutations of the experimentally identified regions. The average expected number of overlaps (*black*) is compared with the observed number of overlaps (*green*). The $\alpha = 0.05$ confidence interval is indicated (*red*). z-scores and permutation test p-values of the differences are also indicated. **c** DRIP-qPCR analyses at the indicated regions in siRNA[dH1] cells, siRNA[lacZ] and untreated cells. Before immunoprecipitation samples were treated with bacterial RNH (+) or not (−) (N = 2). For each position, the fold-change respect to the untreated condition at this position is presented. *Error bars* are s.e.m. The p-values of siRNA[dH1] with respect to siRNA[lacZ] are indicated (no asterisk >0.05, *<0.05, **<0.01, ***<0.005; two-tailed Student's t-test)

associated with R-loops accumulation (Fig. 8a and b). ChIP-qPCR experiments showed that dH1 depletion reduced HP1a occupancy at heterochromatic elements (Fig. 7d), which is consistent with a role in tethering Su(var)3–9 to heterochromatin to regulate H3K9me2,3[15] and, thus, HP1a binding. Conversely, HP1a was not required for dH1 binding to heterochromatin since HP1a depletion did not significantly reduced dH1 occupancy at heterochromatic elements (Fig. 7e). Notably, DRIP-qPCR experiments showed that co-depletion of dH1 in HP1a-depleted cells restored R-loops accumulation in heterochromatin (Fig. 7c). Altogether these results indicate that, in HP1a-depleted cells, relief of heterochromatin silencing per se does not induce R-loops accumulation unless dH1 is also depleted, suggesting that dH1 plays a specific role in preventing R-loops accumulation in heterochromatin.

**dH1 depletion induces JNK-dependent apoptosis.** dH1 depletion in the wing imaginal disc of *his1*[RNAi] flies induced a strong apoptotic wing phenotype[11] (Fig. 8a and b). This phenotype was the consequence of activation of the DNA damage response (DDR) since it was significantly rescued by simultaneous depletion of the main DDR components dATR/mei-41, dATM/Tefu, dChk1/grp and dChk2/Loki (Fig. 8a and b). Notably, expression of human RNH1, which did not affect wing development in wild-type flies (Supplementary Fig. 10),

significantly rescued the wing phenotype of dH1-depleted *his1*[RNAi] flies (Fig. 8a and b), suggesting that DDR induced by dH1 depletion is associated with R-loops accumulation. Interestingly, depletion of dATR/mei-41 resulted in the strongest rescue (Fig. 8a and b) and, under these conditions, γH2Av and activated caspase-3 were undetectable in the wing imaginal disc (Fig. 8c). We also observed that overexpression of the dominant negative p53[H159N] form did not rescue the wing phenotype of dH1-depleted *his1*[RNAi] flies (Fig. 8a and b) and activated caspase-3 levels in the wing imaginal disc remained high (Fig. 8d). Moreover, depletion of p53 had only minor effects (Fig. 8a, b and d). In contrast, inactivation of the JNK-pathway by depletion of dJNK/*basket* or overexpression of the *pucker* phosphatase strongly rescued both the wing phenotype (Fig. 8a and b) and activated caspase-3 levels (Fig. 8e). Altogether, these results suggest that R-loops accumulation induced by dH1 depletion activates DDR and JNK-dependent apoptosis.

## Discussion

Results reported here support a model by which depletion of *Drosophila* linker histone dH1 relieves transcriptional silencing of heterochromatic elements, inducing R-loops accumulation and, ultimately, DNA damage, genomic instability and apoptosis (Fig. 9). Interestingly, in *Caenorhabditis elegans*, derepression of repetitive elements also results in R-loops accumulation[40]. Several

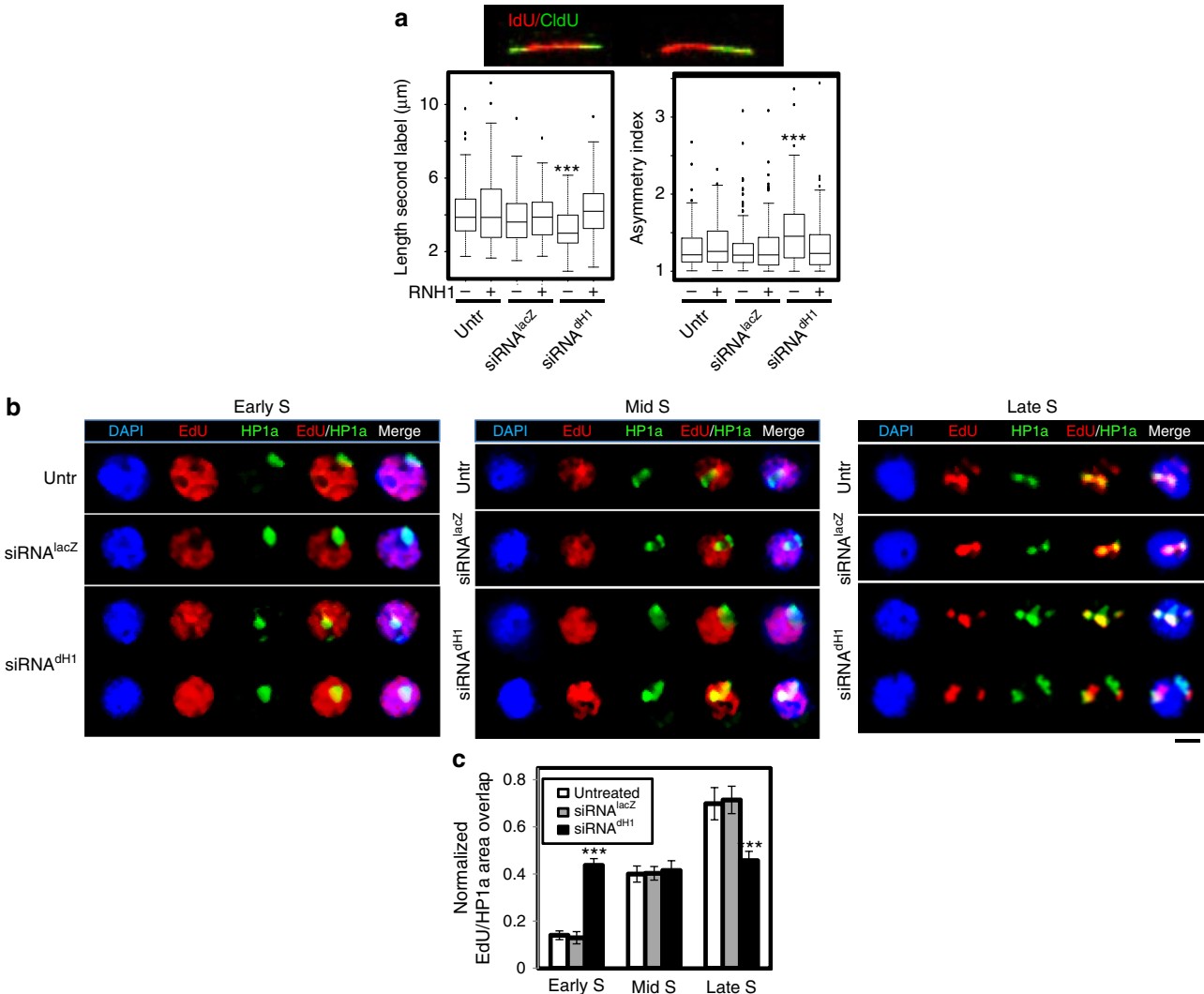

**Fig. 6** dH1 depletion affects DNA replication. **a** On the *top*, DNA fibers labeled with a double pulse of IdU (1st label, *red*) and CldU (2nd label, *green*). On the *bottom*, box-plots showing the length of the 2nd label (*left*) and the asymmetry index (*right*) of DNA fibers obtained from siRNA[dH1], siRNA[lacZ] and untreated cells overexpressing RNH1 ( + ) or not (−) (n > 100 for each condition). *Error bars* are s.e.m. The *p*-values of siRNA[dH1] respect to siRNA[lacZ] are indicated (no asterisk >0.05, ***<0.005; two-tailed Wilcoxon test). **b** Staining for EdU (*red*) and immunostaining with αHP1a (*green*) of siRNA[dH1], siRNA[lacZ] and untreated cells sorted at early, mid and late S-phase. DNA was stained with DAPI (*blue*). *Scale bar* corresponds to 4 μm. **c** The extent of EdU/HP1a colocalization determined as the normalized proportion of total αHP1a area that shows EdU incorporation is presented (n > 50 for each condition). *Error bars* are s.e.m. The *p*-values of siRNA[dH1] respect to siRNA[lacZ] are indicated (no asterisk >0.05, ***<0.005; two-tailed Student's *t*-test)

factors are likely contributing to the accumulation of R-loops that dH1 depletion induces in heterochromatin. On one hand, due to their heterochromatic nature, a high proportion of the transcripts induced by dH1 depletion are probably aberrant and, thus, they are likely processed and exported inefficiently, a circumstance that facilitates R-loops formation[41, 42]. Increased stalling of RNApol II in heterochromatin could also contribute to R-loops accumulation since stalling facilitates R-loops formation[43] and heterochromatin poses a general challenge to transcription. Finally, our results suggest that dH1 plays a specific role in preventing R-loops accumulation in heterochromatin since HP1a depletion, which relieves heterochromatin silencing without affecting dH1 binding, does not induce R-loops accumulation. It is possible that, through the recruitment of specific factors, dH1 regulates R-loops dynamics in heterochromatin. In this regard, dH1 has been shown to co-immunoprecipitate with multiple factors that could prevent R-loops formation or be involved in their resolution. These include several factors associated with RNA and DNA helicase activities, DNA topoisomerases,

and factors involved in RNA processing and transport[44, 45] (Supplementary Table 1). Interestingly, some of these factors are known to localize to heterochromatin and affect heterochromatin silencing[46–48].

Our results indicate that DNA damage induced by dH1 depletion is associated with R-loops accumulation since it is abolished by expression of RNH1, which removes R-loops. The link between R-loops and DNA damage is well established. In particular, several studies suggest that R-loops cause replication fork stalling/collapse, inducing DNA damage[24, 25]. However, while DNA damage induced by dH1 depletion is mainly replicative, abundant R-loops are detected in G1-phase. Several mechanisms could account for the replicative stress caused by R-loops formed during G1-phase (Fig. 9). On one hand, targeting by DNA-damaging agents of the ssDNA in the R-loop has been shown to produce SSBs[49, 50] that are converted into DSBs during DNA replication. Moreover, the nucleotide-excision repair (NER) machinery has been shown to process the DNA:RNA hybrid in the R-loop[51], resulting in ssDNA gaps that provoke DSB during

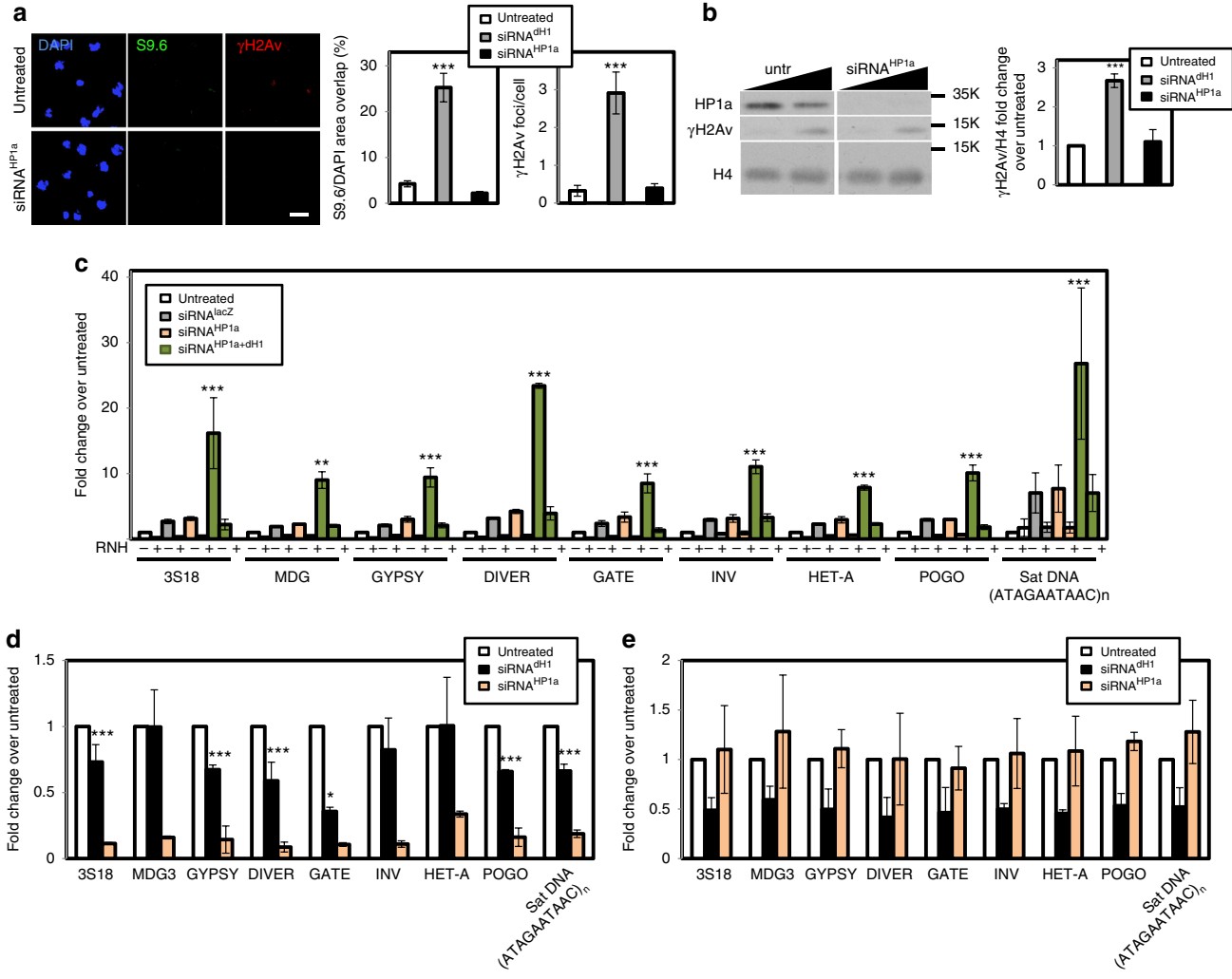

**Fig. 7** HP1a depletion does not induce R-loops accumulation. **a** Immunostainings with αγH2Av (*red*) and S9.6 (*green*) antibodies of siRNA$^{HP1a}$ and control untreated cells. *Scale bar* corresponds to 20 µm. On the *right*, S9.6 (*left*) and γH2Av (*right*) reactivities determined as the proportion of DAPI area stained with S9.6 antibodies and the number of γH2Av foci per cell are presented ($n > 50$ for each condition). The values obtained for siRNA$^{dH1}$ cells are included for comparison. *Error bars* are s.e.m. The *p*-values of siRNA$^{dH1}$ and siRNA$^{HP1a}$ respect to untreated are indicated (no asterisk >0.05, ***<0.005; two-tailed Student's *t*-test). **b** WB analyses with αHP1a, αγH2Av and αH4 antibodies of increasing amounts of extracts (lanes 1–2) prepared from siRNA$^{HP1a}$ and untreated cells. The positions corresponding to molecular weight markers are indicated. On the *right*, quantitative analysis of the results ($N = 2$). The values obtained for siRNA$^{dH1}$ cells are included for comparison. *Error bars* are s.e.m. The *p*-values of siRNA$^{dH1}$ and siRNA$^{HP1a}$ respect to untreated are indicated (no asterisk >0.05, ***<0.005; two-tailed Student's *t*-test). **c** DRIP-qPCR analyses at the indicated repetitive elements in siRNA$^{HP1a}$, siRNA$^{HP1a+dH1}$, siRNA$^{lacZ}$ and untreated cells. Before immunoprecipitation samples were treated with bacterial RNH (+) or not (−) ($N = 2$). For each repetitive element, the fold-change respect to the untreated condition at this repetitive element is presented. *Error bars* are s.e.m. The *p*-values of siRNA$^{HP1a}$ and siRNA$^{HP1a+dH1}$ respect to siRNA$^{lacZ}$ are indicated (no asterisk >0.05, **<0.01, ***<0.005; two-tailed Student's *t*-test). **d** HP1a ChIP-qPCR analysis at the indicated repetitive elements in siRNA$^{HP1a}$, siRNA$^{dH1}$ and untreated cells. For each repetitive element, the fold-change respect to the untreated condition at this repetitive element is presented. *Error bars* are s.e.m. The *p*-values of siRNA$^{dH1}$ respect to siRNA$^{lacZ}$ are indicated (no asterisk >0.05, ***<0.005; two-tailed Student's *t*-test). **e** as in **d** but for dH1 ChIP-qPCR analysis

replication. Finally, taking into consideration the possibility that dH1 is involved in the recruitment of R-loops destabilizing factors, its depletion could significantly extend the relatively short half-life of R-loops[38] to reach S-phase. In this regard, S-phase sorted cells showed intense S9.6 reactivity that, most puzzling, was not significantly reduced upon transient expression of human RNH1 in vivo. However, S9.6 reactivity was significantly decreased, though not abolished, when S-phase sorted cells were treated with bacterial RNH prior to immunostaining, suggesting that, at least in part, it was due to R-loops. It is possible that, in S-phase, RNH1 activity is tightly regulated to control degradation of DNA:RNA hybrids that prime lagging-strand DNA replication. Human RNH1 might miss this regulation in *Drosophila* cells.

The lack of developmental defects associated with RNH1 overexpression in wild-type flies support this possibility since, if targeting DNA:RNA hybrids efficiently in S-phase, RNH1 overexpression must compromise their metabolism during DNA replication and affect development. In addition, significant S9.6 reactivity persisted after treatment of S-phase sorted cells with bacterial RNH treatment in vitro. This residual S9.6 reactivity was strongly reduced to basal levels upon treatment with RNase A, suggesting that S9.6 antibodies, which recognize dsRNAs with low affinity[52, 53], might also be detecting relatively abundant RNA species in S-phase.

dH1 depletion in the wing imaginal disc activates DDR and leads to apoptosis. RNH1 expression rescues apoptosis induced

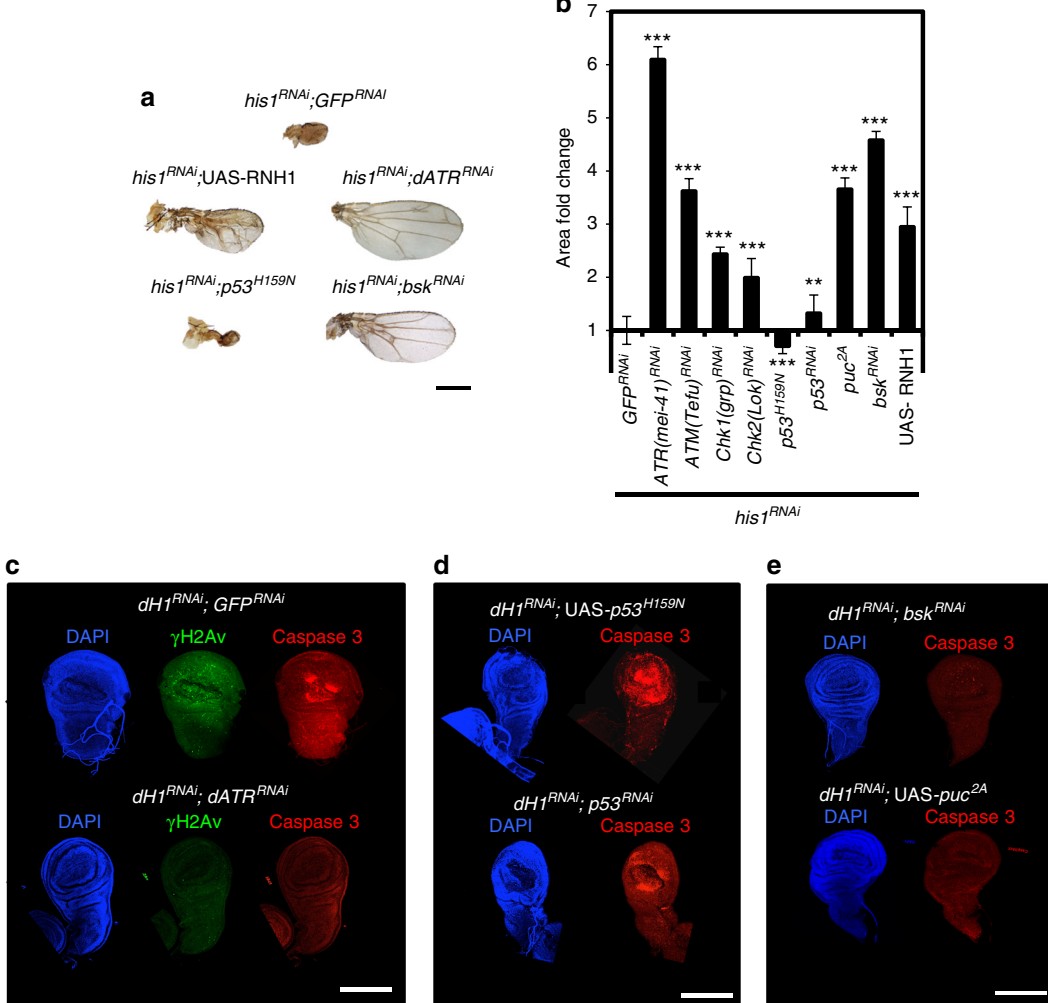

**Fig. 8** R-loops induced by dH1 depletion activate JNK-dependent apoptosis. **a** Wings from dH1-depleted *his1*[RNAi] flies of the indicated genotypes where dH1 depletion was induced in the pouch region of the wing imaginal disc. *Scale bar* corresponds to 500 μm. **b** Quantitative analysis of the wing area of dH1-depleted *his1*[RNAi] flies of the indicated genotypes. Data is expressed as fold change respect to control dH1-depleted *his1*[RNAi]; *GFP*[RNAi] (n > 20 for each condition). *Error bars* are s.e.m. The *p*-values respect to control *his1*[RNAi]; *GFP*[RNAi] are indicated (**<0.01, ***<0.005; two-tailed Student's *t*-test). **c** Immunostaining with αγH2Av (*green*) and αCaspase 3 (*red*) of wing imaginal discs from dH1-depleted *his1*[RNAi] flies upon dATR[RNAi] co-depletion (*bottom*) or not (*top*). DNA was stained with DAPI (*blue*). **d** Immunostaining with αCaspase 3 (*red*) of wing imaginal discs from dH1-depleted *his1*[RNAi] flies upon p53[RNAi] co-depletion (*bottom*) or p53[H159N] overexpression (*top*). DNA was stained with DAPI (*blue*). **e** As in **d** but upon bsk[RNAi] co-depletion (*top*) or puc[2A] overexpression (*bottom*). *Scale bars* in **c**–**e** are 200 μm

by dH1 depletion, indicating that it is associated with R-loops accumulation. From this point of view, our results provide a molecular understanding of the essential role that dH1 plays in cell and organismal viability, suggesting that it is linked to its role in preventing R-loops-induced replicative stress in heterochromatin. Consistent with this hypothesis, DNA damage induced by dH1 depletion is mainly signaled by dATR/mei-41, which is preferentially activated in response to replicative stress[54]. We also show that apoptosis induced by dH1 depletion is JNK-dependent. Interestingly, JNK-dependent apoptosis has been previously reported in *Drosophila* in response to aneuploidy and to limit radiation-induced DNA damage[55, 56].

Our results suggest that, concomitant to R-loops accumulation, dH1 depletion induces premature heterochromatin replication in early S-phase. Notably, in *Physarum*, H1 depletion alters replication timing, shifting late replicating regions into early replication[57]. Interestingly, R-loops have been shown to efficiently prime DNA replication[58]. Thus, it is tempting to speculate that their accumulation in heterochromatin could

induce unscheduled initiation resulting in early heterochromatin replication. Our results also suggest that R-loops induced by dH1 depletion slow down DNA replication globally. Whether this effect reflects direct collisions between incoming replication forks and R-loops-induced lesions remains unclear since no general R-loops accumulation, or γH2Av increase, was detected in dH1-depleted cells. In this regard, it is also possible that the effects on global DNA replication are an indirect consequence of checkpoint activation, and/or uncoordinated initiation of DNA replication[54, 59], caused by the abnormal R-loops accumulation and DNA damage induced by dH1 depletion in heterochromatin. At this stage, a better understanding of the contribution of dH1 to the general pattern of DNA replication, and its dynamics during replication, appears essential.

The contribution of dH1 to R-loops dynamics and maintenance of genome stability are likely conserved. In *S. cerevisiae*, deletion of the H1-like gene *Hho1* results in genomic instability and hyperrecombination in the highly repetitive rDNA locus[60], a defect that, among several possibilities, could arise from

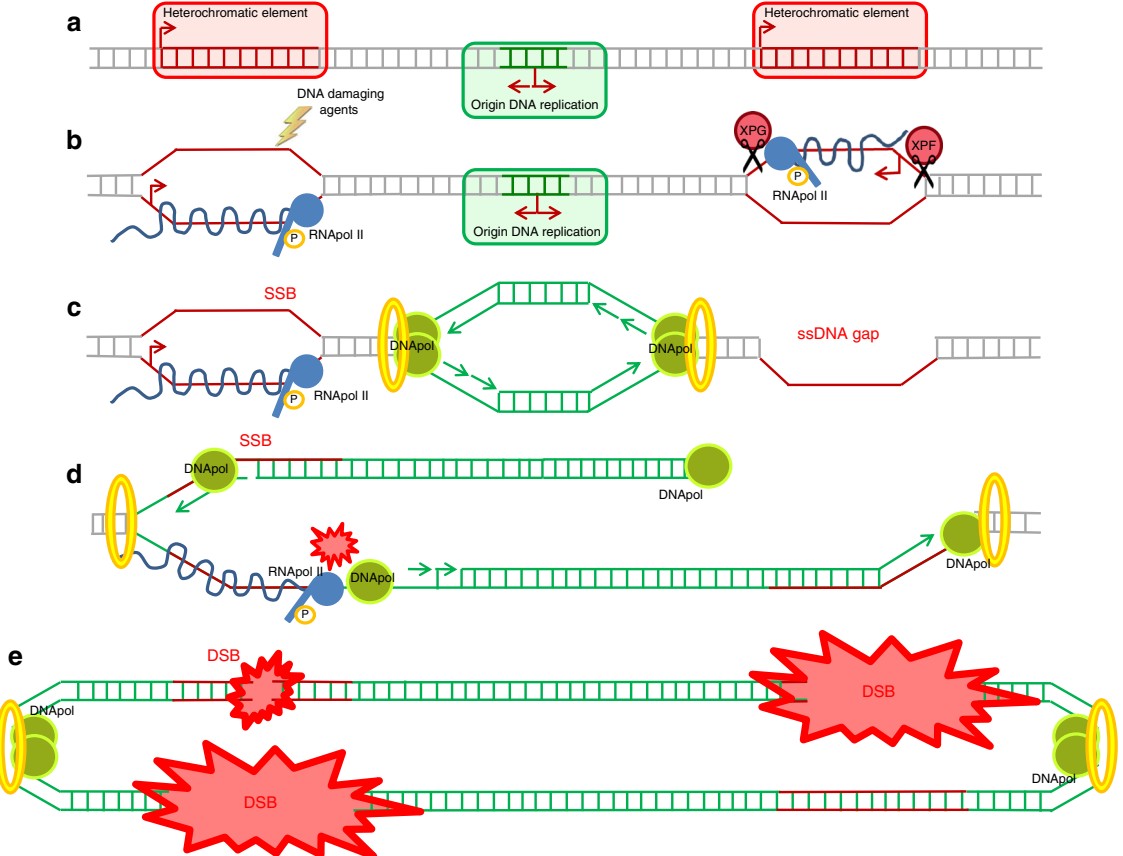

**Fig. 9** Model for the contribution of dH1 to maintenance of genome integrity. In the absence of dH1, expression of heterochromatic elements is upregulated **a**, inducing the accumulation of R-loops **b** that could generate SSBs through targeting of the ssDNA regions by DNA-damaging agents or ssDNA gaps through processing of the DNA:RNA hybrids by the NER system **b** and **c**. During DNA replication, SSBs and ssDNA gaps are converted into DSBs **d** and **e**. Collisions with the replication machinery could also stall replication fork progression and generate DSBs **d** and **e**

increased R-loops formation. Furthermore, in *Drosophila*, expression of several human H1 variants rescues the apoptotic wing phenotype associated with dH1 depletion in the wing imaginal disc[11], suggesting that human H1s can also prevent R-loops accumulation. In summary, our results unveil a novel and essential contribution of linker histones H1 to R-loops dynamics in heterochromatin and, thus, to maintenance of genome integrity and stability, a function that is highly compromised in cancer and is central for cell and organismal viability.

## Methods

**Antibodies**. Mouse monoclonal S9.6, rabbit polyclonal αCID and rat polyclonal αHP1a are described in[34, 61, 62]. Rabbit polyclonal αdH1 was kindly provided by Dr Kadonaga and is described in ref. [11]. All other antibodies used in these experiments were commercially available: monoclonal mouse αγH2Av (DSHB, UNC93-5.2.1) or rabbit αγH2Av (Rockland, 600-401-914), rabbit polyclonal αH2Av (Active Motif, 39716), rabbit polyclonal αH4 (Abcam Ab10158), rabbit polyclonal αCaspase 3 (Cell Signalling, 9661S), mouse polyclonal α−βtubulin (EMD-Millipore, MAb3408), rat monoclonal αBrDU, (AbD Serotec, OBT0030G) and mouse monoclonal αBrDU (Becton Dickinson, 347580). Commercial secondary antibodies used for immunofluorescence were either coupled to cyanines (Jackson) or to Alexa fluorophores (ThermoFisher), and to horseradish peroxidase for WB (Jackson).

**Fly stocks and procedures**. *Tefu*(*dATM*)[RNAi] (V22502), *mei-41*(*dATR*)[RNAi] (V103624), *grapes*(*dChk1*)[RNAi] (V12680), *Loki* (*dChk2*)[RNAi] (V110342), *dp53*[RNAi] (V38235) and *basket*(*dJNK*)[RNAi] (V104569) lines were obtained from the Vienna *Drosophila* Resource Center (VDRC), and UAS-*p53*[H159N] (BL8420) line from the Bloomington Stock Center. *his1*[RNAi] (31617R-3) line was obtained from NIG-FLY and is described in ref. [11]. *GFP*[RNAi] and *hp1a*[RNAi] lines are described in ref. [61]. *nub*-GAL4 and UAS-*Dic2* were kindly provided by Dr. Casali and are described in

Bloomington Stock Center. UAS-*puc*[2A] was kindly provided by Dr Martín-Blanco. UAS-RNH1 (human) line was prepared by subcloning the corresponding cDNA from pcDNA3-RNH1[63] into pUAST, and injected for random integration into w[1118] embryos.

dH1 depletion was induced in the pouch region of the wing imaginal discs and in polytene chromosomes using *his1*[RNAi]; *nub*-GAL4; UAS-*Dic2* flies by themselves or combined with either additional RNAi lines or overexpression constructs. Similarly, HP1a depletion was induced in the same tissues using *hp1a*[RNAi]; *nub*-GAL; UAS-*Dic2* flies. To analyze wing phenotypes, wings were prepared and recorded as described[11]. Wing images were measured and analyzed using FIJI software[64].

**Knockdown experiments in S2 cells**. For dH1 knockdown experiments, *Drosophila* S2 cells (ATCC CRL-1963) were grown to $5 \times 10^5$ cell mL$^{-1}$ and incubated for 8 days with 100 µg of dsRNA against dH1 or against LacZ (as a control) added to the culture at days 1, 4 and 7. For HP1a knockdown experiments, cells were incubated for 6 days with 30 µg of dsRNA against HP1a added to the culture at days 1 and 3. For HP1a + dH1 co-depletion, cells were incubated for 8 days with 100 µg of dsRNA against dH1 at days 1, 4 and 7 and with 30 µg of dsRNA against HP1a at days 5 and 7. dsRNA was prepared using MEGAscript T7 Transcription Kit (Ambion) and the primers indicated in Supplementary Table 2. The efficiency of depletion was determined by Western blot (WB) analyses using rabbit polyclonal αdH1 (1 : 20,000), rabbit polyclonal αH4 (1 : 2000) and rat polyclonal αHP1a (1 : 10,000) antibodies. The effects on H2Av and γH2Av were determined by WBs using αH2Av (1 : 2000) and αγH2Av (1 : 2000) antibodies. Uncropped WBs are shown in Supplementary Fig. 11.

**Single-cell electrophoretic analyses**. For single-cell electrophoretic analyses, cells were placed on slides and embedded into a 0.5% low-melting agarose matrix, lysed overnight at 37 °C in neutral buffer (0.5 M EDTA, 0.5 mg mL$^{-1}$ proteinase K, 2% sarkosyl, pH 8) or at 4 °C in alkaline buffer (2.5 M NaCl, 100 mM EDTA, 10 mM Trizma base, 1% sarkosyl, 1% Triton X-100, pH 10), and subjected to electrophoresis at 3 or 13 mA, respectively. After electrophoresis, slides were stained with 10 µg mL$^{-1}$ Hoechst (Sigma) to visualize DNA. Images were recorded on a Nikon E1000 fluorescence microscope and processed using CASP software[65].

**SCE experiments**. For SCE experiments, cells were cultured in media containing 20 µg mL$^{-1}$ BrdU (Sigma) for 48 h. Then cells were collected and incubated in hypotonic solution (50 mM glycerol, 5 mM KCl, 10 mM NaCl, 0.8 mM CaCl$_2$, 10 mM sucrose) for 5 min, fixed in methanol:acetic acid (3:1, v/v) and placed drop-wise on slides. After drying, slides were incubated with 0.5 µg mL$^{-1}$ Hoechst (Sigma), subjected to UV irradiation for 1 h, incubated in SSC (0.3 M sodium chloride, 0.03 M sodium citrate) at 62.5 °C for 2 h and stained with 20% Giemsa in Sorensen's buffer (0.133 M Na$_2$HPO$_4$, 0.133 M KH$_2$PO$_4$, pH 7) for 25 min at room temperature and mounted in Permount media.

**X-ray irradiation**. For X-ray irradiation, cells were subjected to 10 Gy in a Xylon SMART200 apparatus at room temperature. Aliquots were taken at specific time points and analyzed by WB using rabbit polyclonal αγH2Av (1 : 2000) and mouse polyclonal α-βtubulin (1:10,000). Uncropped WBs are shown in Supplementary Fig. 11.

**Cordycepin treatment**. For inhibition of transcription, S2 cells were incubated with 200 µM cordycepin (Sigma-Aldrich) for 0, 2 and 4 h. After incubation, cells were processed for immunostaining as indicated below.

**Immunostaining Experiments**. For Immunostaining experiments, cells were attached to slides pre-treated with concanavalin A, fixed in 4% *p*-formaldehyde, washed in PBS, permeabilized in PBST (PBS, 0.3% Triton), blocked in PBSTB (PBS, 0.3% Triton, 0.2% BSA) and incubated with primary antibody diluted in PBSTB. After incubation, cells were washed in the same solution, incubated with secondary antibody diluted 1 : 400 in PBSTB, washed again and mounted in DAPI-MOWIOL. For sorted cells, immobilization was performed immediately after sorting onto a slide using cytospeen and, then, cells were immunostained as described above. For immunostaining with S9.6 antibody, cells were incubated with 0.5% SDS in PBS prior to PBST treatment[63]. Antibody dilutions used were: rabbit polyclonal αγH2Av (1 : 500), mouse monoclonal S9.6 (1 : 500), and rat polyclonal αHP1a (1:400) and αdH1 (1:4000). Images were recorded on Leica TCS SPE and Leica TCS SP2 AOBS system confocal microscopes, and analyzed using FIJI software[64]. S9.6 immunostaining was filtered for DAPI colocalization. Briefly, for each nucleus, 0.5 µm thick confocal stacks were recorded from bottom to top in a Leica SPE confocal microscope. Stacks were then processed by filtering 8-bit images for S9.6 immunostaining with the corresponding 8-bit DAPI images to subtract S9.6 signals not colocalizing with DAPI and converted to RGB. For quantification, maximal projections of the RGB stacks were obtained and the total S9.6 area was determined and expressed as percentage of total DAPI area. Overlaps between γH2Av, S9.6, EdU and HP1a signals were determined on maximal RGB projections of the corresponding 0.5 µm thick confocal stacks and expressed as the proportion of area overlap. Statistical analysis of γH2Av, S9.6 and EdU colocalization with HP1a was performed in R version 2.15.3 fitting a generalized linear model and adjusting by total HP1a area.

For immunostaining in imaginal discs and polytene chromosomes, third instar larvae were prepared and immunostained[61] with mouse polyclonal αγH2Av (1 : 100), rabbit polyclonal αCaspase 3 activated (1 : 100), rabbit polyclonal αdH1 (1 : 4000), mouse polyclonal αp53 (1 : 100), mouse monoclonal S9.6 (1 : 500) and rat polyclonal αHP1a (1 : 500). For S9.6 immunostaining, polytene chromosomes were incubated for 5 min with 0.5% SDS in PBS after fixation and prior to blocking. Images were recorded on Leica TCS SPE and Leica TCS SP2 AOBS system confocal microscopes, and analyzed using FIJI software[64]. S9.6 immunostaining was filtered for DAPI colocalization as described above.

**FACS sorting**. For FACS sorting, cells were fixed with ethanol and stained with 1 µg mL$^{-1}$ DAPI (Sigma). The minimal quantity of S-phase cells sorted was established at 10$^6$ cells for WB analyses and 10$^5$ for immunostaining experiments. For sorting into early, mid and late S-phase, the S-phase was subdivided in three equally wide gates. Approximately 10$^5$ cells for each S-subphase were sorted and used for immunostaining experiments. When indicated, sorted cells were treated before immunostaining with 2 µL (4U) of recombinant bacterial RNase H (RNH) (Invitrogen) alone or together with DNase-free RNase A at a final concentration of 0.2 mg mL$^{-1}$ in PBS1X-0.1% Triton X-100 for 30 min at 37 °C.

**Transient RNase H1 (RNH1) expression in S2 cells**. For experiments in which human RNH1 was transiently expressed, S2 cells were transfected with either 3 µg of plasmid pcDNA3-RNH1 expressing RNH1[63] or the corresponding pcDNA3 empty plasmid (as control) using calcium phosphate precipitates. Cells were collected 48 h after transfection.

**EdU and double IdU+CldU labeling experiments**. For EdU (5-ethynyl-2′-deoxy-yuridine) incorporation experiments, exponentially growing S2 cells were incubated with 10 µM EdU (ThermoFisher) for 10 min at 25 °C. Subsequent EdU click-it detection (ThermoFisher Scientific) was performed following manufacturer's protocols. Briefly, detection is based on a copper-catalyzed covalent reaction between an azide (provided by EdU) and an alkyne (provided by the Alexa Fluor dye). To determine the extent of EdU incorporation in heterochromatin, 0.5

µm thick confocal slices were processed by filtering 8-bit images for HP1a immunostaining with the corresponding 8-bit EdU images, maximal projections were obtained, merged, converted to RGB, and quantified as described above.

For double IdU and CldU pulse labeling experiments, 2–3x10$^6$ cells were supplemented with 40 µM IdU (Sigma I7125) from a 2.5 mM IdU stock in Schneider's medium, and incubated for 20 min at 25 °C. After incubation, cells were washed with Schneider's medium and incubated for additional 20 min at 25 °C with Schneider's medium containing 40 µM CldU (Sigma C6891). After labeling, cells were collected, washed in PBS and resuspended at 10$^6$ cells mL$^{-1}$ in PBS. Then, cells (in a 2 µL drop) were placed on slides, let dry for 7 min at room temperature and incubated for 2 min at room temperature with 7 µL of spreading buffer (200 mM Tris-HCl, pH 7.4, 50 mM EDTA, 0.5% SDS). After incubation, slides were tilted about 15–30° and drops were let to run down slowly for 2–5 min. Samples were air-dried and fixed with acetic acid: methanol (3 : 1, v/v) for 10 min and stored at −20 °C until staining. Prior to immunostaining, slides were washed twice in distilled water for 5 min, briefly rinsed in 2.5 M HCl and denatured in the same solution for 1 h, washed and incubated in blocking solution (PBS, 1% BSA, 0.1% Tween20) for 30 min$^{-1}$ h. Then, samples were incubated overnight with rat monoclonal αBrdU (1 : 200) (AbD Serotec), which detects CldU, and mouse monoclonal αBrdU (1 : 200) (Becton Dickinson), which detects IdU, in blocking buffer. Next day, samples were washed with PBS, fixed in 4% *p*-formaldehyde for 10 min, washed, incubated with α–rat Alexa fluor 488 and α–mouse Alexa fluor 555 (1 : 400) in blocking buffer, washed and mounted in MOWIOL. Images were taken with a Leica SP2 confocal microscope and DNA fibers were measured using FIJI software[64]. The length of the second label (green) in double-labeled fibers was measured for > 100 well-isolated fibers for each condition. Asymmetry index was calculated as described in[24, 39], considering only double-labeled fibers. The lengths of the two labels were measured and for each individual fiber the ratios between them (red/green and green/red) were calculated and those above unit were selected. ~ 100 well-isolated fibers were analyzed for each condition.

**ChIP experiments**. For ChIP experiments, chromatin was prepared from S2 cells, sonicated to obtain fragments ranging from 200 to 500 bp and immunoprecipitated (IP) using 3 µL of rabbit polyclonal αγH2Av, 1 µL of rabbit polyclonal αdH1 or 1 µL of rat polyclonal αHP1a. Immunoprecipitated DNA was eluted and purified according to standard protocols.

For γH2Av ChIP-seq, two independent biological replicates for each condition were analyzed. Library construction, cluster generation and sequencing analysis were performed following manufacturer's protocols (www.illumina.com). Reads were aligned against the dm3 UCSC genome 2012 release using Bowtie 0.12.5, allowing two mismatches in the read seed and considering all possible alignment sites for each read in both the immunoprecipitated (IP) and input samples (setting an upper limit of 10.000 to limit output file size), in order to compare enrichment of genomic features with multiple copies across the genome (i.e. transposons). Assessment of the immunoprecipitation was done with SSD metrics and PCA-type plots using the htSeqTools package version 1.20[66]. Further quality control information was performed with the FastQC software version 0.10. Identification and removal of potential PCR-amplification artifacts was performed with sambamba v0.5.1 using the markdup option and default settings. Coverage TDF tracks were generated with the IGV software version 2.1.16. Peak calling between IPs and their respective input in untreated and dH1-depleted cells was performed with MACS 1.4 using options –g dm and leaving the rest as default (*p*-value <1e-05). The DiffBind version 1.4.2 was used with the resulting called peaks to further determine differentially enriched or depleted regions between untreated and dH1-depleted conditions using the edgeR method[67]. All identified sites showing |FC| > 1.15 were annotated for overlapping and closest genes against the Ensembl genome annotation version 71 (April 2013) using the annotatePeakInBatch function from the ChIPPeakAnno package[68]. version 2.6.1. FASTA sequences for differentially enriched/depleted sites were retrieved with the getSeq function from the Biostrings package v2.26.3 (using the annotation package BSgenome.Dmelanogaster.UCSC.dm3 v1.3.19) and were analyzed with the RepeatMasker software (http://www.repeatmasker.org, version open-4.0.5)[69] in order to identify repeated elements found among them. Peak location and coverage plots were generated with the htSeqTools package version 1.20.0. Permutation tests to assess enrichment of repeated elements among identified peaks were performed with the regioneR package version 1.2.3[70], using the UCSC dm3 RepeatMasker track (March 2017).

For ChIP-qPCR, triplicates from two independent biological replicates were subjected to SYBR Green I-based real-time PCR using LightCycler 480 SYBR Green I Master Mix in a LightCycler® 480 Instrument (Roche). The percentages of immunoprecipitated material were calculated by the ΔΔCt method and expressed as fold change over the control untreated condition. Primers used in these experiments are described in Supplementary Table 2.

**DRIP experiments**. For DRIP experiments, genomic DNA was purified, treated with 1 mg mL$^{-1}$ RNase A, sonicated to 500bp–1000bp fragments, treated or not with 3 µL (6 U) of RNase H (RNH) (Invitrogen), and subjected to IP with 4 µg of mouse monoclonal S9.6 antibody. Immunoprecipitated DNA was eluted and purified using standard protocols.

For DRIP-seq, two independent biological replicates for each condition were analyzed. Library construction, cluster generation and sequencing analysis were performed following manufacture's protocols (www.illumina.com). DRIP-seq data were aligned and processed exactly as done for ChIP-seq. To determine location of R-loops across the whole genome, we first performed a peak calling using MACS 1.4 in RNH untreated samples using the same options as for the ChIP-seq analysis. Then, $\log_2$ RPKM immunoprecipitated signal over identified peaks was measured for both RNH untreated and treated sample pairs, and only those locations showing at least 30% reduction upon RNH1 treatment in both replicates were kept (FC < −1.5). Finally, these results were used with the DiffBind package 1.4.2 in order to determine R-loops regions specifically enriched/depleted in dH1-depleted cells using the edgeR method, as described above for ChIP-seq analysis. Site annotation, random permutation tests, RepeatMasker analysis and coverage plots were performed using the same procedures as for the ChIP-seq data.

For DRIP-qPCR, triplicates from two independent biological replicates were subjected to real-time PCR using LightCycler 480 SYBR Green I Master Mix in a LightCycler 480 Instrument (Roche). The percentages of immunoprecipitated material was calculated by ΔΔCt method and expressed as fold change over the control untreated condition not treated with RNH1. Primers used in these experiments are described in Supplementary Table 2.

**RT-qPCR experiments**. Total RNA was purified using Qiagen's RNeasy Midi Kit according to manufacturer's guidelines. Briefly, nucleic acids were extracted using Trizol reagent, digested with DNAse I, and purified. Then, 1 μg of total RNA was subjected to reverse transcription with random primers using Roche's Transcriptor First Strand cDNA Synthesis Kit and cDNA was subjected to real-time PCR analyses using LightCycler 480 SYBR Green I Master Mix in a LightCycler 480 Instrument (Roche). Relative expression levels quantifications were calculated by ΔΔCt method and normalized respect to *actin5C*. Results were expressed as fold change over untreated condition. Primers used in these experiments are described in Supplementary Table 2.

**Data availability**. γH2Av ChIP-seq and DRIP-seq data are deposited at NCBI GEO (GSE99016). All relevant data are available from the authors. Requests for materials should be addressed to Jordi Bernués (jordi.bernues@ibmb.csic.es).

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

## Acknowledgements

We are thankful to Drs A. Casali, J. Casanova, O. Fernández-Capetillo, G. Fernández-Miranda, J.T. Kadonaga, R. Méndez, G. Roncador, T. Stracker and A. Vaquero for materials and advise. This work was supported by grants from MICINN (BFU2012-30724 and BFU2015-65082-P), the Generalitat de Catalunya (SGR2014-204) and by the European Community FEDER program. A.B.-F. and A.C.-L. acknowledge receipt of FPU (MED) and FPI (MINECO) fellowships, respectively.

## Author contributions

A.B.-F., A.C.-L. and J.B. performed the experiments. O.R. performed the Bioinformatics analyses. J.B. and F.A. designed the experiments and wrote the paper.

## Additional information

**Competing interests:** The authors declare no competing financial interests.

