## [Peer Review File · Nature Communications]

Reviewers' Comments:

Reviewer #1 (Remarks to the Author)

Complete or even partial loss of function of the linker histone H1 induces lethality in both mice and *Drosophila*. Bayona et al. exploit the existence of a single variant of histone H1 in *Drosophila*, with the intent of elucidating its essential function. The authors report that knock-down of H1 induces an S-phase specific DNA damage response ATR-dependent. Cells tend to accumulate in S-phase, display delayed fork progression and altered replication timing. The authors indicate that γ H2Av, the marker used to follow the DNA damage response, tends to accumulate at repetitive sequences. Knock-down of histone H1 also induces the up-regulation of the expression of these same genomic regions. Thus the authors hypothesize that the DNA damage response could be induced by the accumulation of R-loops formed as a consequence of increased transcription. Indeed, the authors could detect an accumulation of R-loops upon knock-down of H1, at least during G1. However, importantly, the authors show that transcription per se is not sufficient to induce the accumulation of R-loops. Indeed, knock-down of HP1 also up-regulates transcription of the repetitive sequence, but does not increase R-loop formation. This result suggests that H1 is specifically involved in the resolution of R-loops metabolism.

The topic is important and the results interesting. In principle the experiments are all there, but the results are sometimes incomplete, lacking controls and definitely there is a general lack of explanation of the statistics employed. The text is also presented in an order that I found hard to follow logically and it would benefit from being turned around. At the present state the paper cannot be published as it is, but it should be revised. Below follow some major points.

Extended Data Figure 1: a. Since the positivity for γ H2Av is scored on the number of foci per cell and not as number of cells positive for γ H2Av, insets showing the foci should be shown for both control and knock-down.

b. In all the western blots of γ H2Av, throughout the paper, the loading control used is always histone H4. H2Av should really be used. It is crucial to ensure that no increase in the basal level of the histone is taking place.

Extended Data Figure 3: a. It would help if some graphic reference like centromere and telomeres would be indicated.

d. I could not find anywhere how the average expression has been calculated to then obtain the list of genes with an expression higher than average.

Figure 1: b and c. In general, in all the ChIP data displayed the scale of the y axis is missing. Although I am sure it is consistent within one figure, it would be very important in order to compare data across different figures. Looking at b. and also at Extended Data Figure 3b, the impression is that there is a general increase in the background of γ H2Av, and it is not so different between non-repetitive (Ext. Data Fig. 3b) and repetitive (Fig. 1b) regions.

d. This data are not convincing. HP1a has a very wide localization. As the number of γ H2Av foci is increased upon H1 knock-down, the probability of HP1a- γ H2Av co-localization by chance would be higher than in the control cells. This has to be sorted statistically and by using irradiated cells as control. In addition, is HP1a covered nuclear area comparable H1-knocked-down cells and controls? In this figure HP1 looks more extended in the knocked-down cells.

e and f. In both cases the resolution is too low to claim that γ H2Av is somewhere specific. In e. FISH-IF could be used.

Figure 2: a. This figure should be removed. Since it is shown in Extended Data Figure 6 that H1 knock-down causes an accumulation of cells in S-phase and according to Figure 3a the R-loops are much higher in S-phase irrespective of H1 depletion, concluding from cycling cells that R-loops are higher upon H1 knock-down is wrong. This conclusion can be reached correctly as in Figure 3a.

b. Here there is somehow the same problem as in Figure 1d. In the statistics to calculate the significance was it taken into consideration the probability of random co-localization due to the extent of HP1a staining and the increase in the number of S9.6 signals? In any case, the same figure for untreated cells should be shown.

Extended Data Figure 6: a. It seems that in the control cells EdU and γ H2Av also co-localize. However the picture is too small and in b. the same magnification should be shown for the control.

d. There is no information of the statistics and the number of samples used for this figure. The differences are so minimal that it is important to explain how the statistical significance was obtained.

Figure 3: a. The increase of γ H2Av between G1 and S-phase seems to be very similar between the control and H1 knock-down. Again, since there is no information of the statistics, I assume that the 3 asterisks refer to the difference between control and knock-down.

d. Presenting only the overly complicates the visualization. In addition, it would be much more informative to show a classification of replication patterns in the control and the corresponding (or not-corresponding) patterns in the si-RNA treated cells.

Figure 4: e. In the figure provided the levels of γ H2Av between untreated cells and cells treated with the si-RNA against HP1a look the same.

Finally, the sentence stating, " Altogether, these results support a model by which dH1 depletion relieves transcriptional silencing of heterochromatin regions, favoring R-loop FORMATION" should be changed. There is no evidence for an effect on formation. May be resolution? However, the only fair conclusion is that H1 is involved in R-loop metabolism.

Reviewer #2 (Remarks to the Author)

Bayona and colleagues report in this manuscript that linker histone H1 serves to suppress cryptic transcription at repeated heterochromatic sequences including transposons. Their evidence shows that H1 depletion leads to reactivation of these transposons and to the formation of γ H2Av foci, indicative of DNA damage. Through a series of experiments involving microscopy, genetic manipulation, ChIP, and analysis of RNA:DNA hybrid and replication patterns, the authors suggest an interesting mechanism by which deregulated transcription leads to R-loop formation which in turns leads to early DNA replication and DNA damage. The hypothesis is interesting but the manuscript stills feels premature at that point and certain key points are not sufficiently fleshed out or supported by the data.

One of the important first bit of data regards the identification of H1-specific H2Av loci using ChIP-seq. According to Extended Fig 3, the authors have identified on the order of 15 such loci genome-wide. This seems extremely low and puts the relevance of the findings in question. This section also suffers from a lack of information. What was the size of the peaks reported? Readers must get an idea of the resolution achieved here. The statement that H1-independent peaks tend to map to TSS puzzles me a little as H2Av tracks, at least in mammalian systems, are very large (>100 kb) and can't be compared to TSS. It is also unclear how many independent biological replicates of the ChIP-seq were performed, including input? How did authors deal with problems of background subtraction and normalization over the repetitive portion of the genome? Clarifying and expanding this section is critical to establish the relationship between linker H1 depletion and γ H2Av induction.

Another key argument put forth by the authors is that H1 depletion leads to R-loop accumulation. This is perhaps not entirely surprising but given the limited number of tools available to study R-loops, the data is not always convincing. One question raised here regards the amount of R-loops induced upon H1 depletion. The authors use microscopy for this and report the data as S9.6/DAPI (%area). In Fig 2a this number reaches 20% in dH1-depleted cells with a nice and small error bar. In Ext Fig 5 however, this number is now 40% with a similarly small error bar. Why the discrepancy? Is this saying that the method is in fact not quantitative? Regarding DRIP qPCR data, I would much prefer if the authors presented the data as % input on the y-axis. This bad habit of normalizing the data to a control completely prevents readers from gauging the frequency at which these structures may form. This is important information.

The authors also seem bent on showing that "that S9.6 foci significantly colocalize with HP1a in dH1-depleted cells". Yet, the data actually shows that only a very small portion of S9.6 signal colocalizes with HP1a based on Figure 2b and 2c. In conditions of H1 depletion, this colocalization may increase a bit but it remains marginal. I urge the authors to tone down their interpretations to better reflect reality.

Finally, perhaps the most confusing part of the manuscript regards the RNase H sensitivity (or lack thereof) of the S9.6 signal. Throughout much of Figures 2 and 3, the authors use RNASEH1 expression to show that the R-loop signal is "real" because it is sensitive to RNase H activity. This is pretty much standard operating procedure in the field despite accumulating evidence that RNASEH1 expression is very toxic to cells, and may cause both transcriptional responses and impair DNA replication. I would urge the authors to study possible effects of RNASEH1 expression on DNA replication more carefully as it is possible that the effects they are documenting are caused indirectly via a reduction in DNA replication caused by RNASEH1. Then of course, when authors analyzed various phases of the cell cycle, the S9.6 signal was variably sensitive to RNASEH1, with S and G2/M phases showing RNASEH1-insensitive signal. This is extremely confusing and hard to fit in with any mechanism. The suggestion by the authors that S9.6 recognizes RNA primers is doubtful and the reference 18 provides no evidence whatsoever for that claim - it must be deleted. If the authors want to claim that S9.6 recognizes Okazaki primers, they should provide direct evidence. Of course S9.6 has a documented minor affinity for AU-rich dsRNA but it is a bit unclear why such reactivity should obscure the signal only in S and G2/M. In short it seems that the authors don't understand what's happening there more than I do and this is worrisome.

Other important points:

- Often through the manuscript it is quite unclear how many independent experiments were performed or how many cells were analyzed to reach the conclusions reached by the authors. Similarly, some key technical details are sometimes missing. For ex. In Fig 1a, how many cells were analyzed? I would expect $n > 50$ to reach any kind of statistical confidence. In Extended Figure 2, how many times was this repeated? The absence of error bars leads me to think this was done only once. In Fig 2a and 2b, how many cells were analyzed?
- In Fig 1d, the supposed co-localization between H2Av and HP1a is far from clear on the image provided.
- In Fig 1e, the "intense γ H2Ac reactivity observed at pericentric and telomeric heterochromatin in dH1-depleted chromosomes" seems at odd with the data in Fig 1d. Is that a property of metaphase chromosomes? This is confusing.
- In Fig 1f, the statement that increased SCE occur "preferentially at centromeric and telomeric regions" isn't supported. Provide data or delete.
- The statement that "55% of γ H2Av foci detected in dH1-depleted cells colocalize with S9.6 foci" isn't supported. Provide data or delete.
- Fig 4 reports interesting genetic data using the wing model but it feels a little disconnect from the rest of the study. Likewise the observation of JNK mediated apoptosis is interesting but should also be studied in S2 cells.
 - In their model, the authors assume that firing origins in heterochromatin are more likely to break if they encounter R-loops. However, their data also show that heterochromatic regions shift to an earlier replication time in H1-depleted cells. This suggests that R-loops, directly or not, may favor origin firing. As the authors cite, there is some interesting evidence that R-loops might be able to prime DNA synthesis. This should be added.

Minor points:

- DNA breaks are referred to as DBs or DSBs, please be consistent.
- In Figure 2d, isn't the GATE LTR element upregulated in siRNA LacZ conditions?

Reviewer #3 (Remarks to the Author)

In this manuscript, Bayona et al., have investigated the role of the linker histone H1 in preventing genomic instability. They find that in absence of H1, there is a substantial increase in DNA lesions arising from heterochromatic structures mainly transposon elements. They also find that these lesions that due to increased transcription of the heterochromatic elements that lead to increased R loop formation in G1 that collide with the replication machinery in S phase.

The manuscript addresses an interesting question with a substantial set of experiments.

I have here some points that will further strengthen the manuscript and its conclusion

1. The Immunofluorescence images are in the most of the cases not very convincing. For example in Figure 1D at the siRNA^{dH1} condition there is no apparent increase in the overlap of γ -H2AX and the HP1 domain and the overlap might be due to the obvious increase of the HP1 domain possibly by decondensation. The same stands for figure 2B. Moreover images of 2A, 3A, 4D are of very low magnification and higher magnifications should be provided.
2. The main biological outcomes of the depletion of H1 like the sister chromatid exchange depicted have to be tested in the presence of RNAase H to conclude that this phenotype is really the consequence of increased R loop formation in the absence of dH1.
3. Minor spelling mistakes include page 2 line 20 DSBs instead of DBs and page 4 line 8 increase instead of increased.

REVIEWER #1**General comment**

The referee thinks that "***there is a general lack of explanation of the statistics employed***". In this revised version we describe in the **Methods** section the statistical analyses we have used in detail and full information is provided in the **Figure Legends**, indicating in each case the number of independent experiments carried out, the sizes of the samples analyzed and the statistical tests used.

Major points

1) Extended Data Figure 1: a. Since the positivity for γ H2Av is scored on the number of foci per cell and not as number of cells positive for γ H2Av, insets showing the foci should be shown for both control and knock-down.

Enlarged images of representative individual cells showing γ H2Av foci have been incorporated to **Figure 1a** (previously Extended Data Figure 1a).

b. In all the western blots of γ H2Av, throughout the paper, the loading control used is always histone H4. H2Av should really be used. It is crucial to ensure that no increase in the basal level of the histone is taking place.

We fully agree with the reviewer that it is important to show that the basal levels of H2Av are not affected upon dH1 depletion. However, simultaneous detection of H2Av and γ H2Av by WB proved to be difficult in our hands. Instead, we checked in several independent experiments the levels of H2Av in dH1-depleted cells, finding that they were unaffected respect to control undepleted cells. These results are now presented in **Supplementary Figure 1** and in the **Results** section (page 4, lines 10 and 11). We thank the reviewer for pointing out this important control that was missing in our previous version of the manuscript.

2) Extended Data Figure 3: a. It would help if some graphic reference like centromere and telomeres would be indicated.

In this revised version we describe in the **Figure Legend** corresponding to **Figure 2a**, the nomenclature used to indicate the different assembled and unassembled *Drosophila* chromosomes according to dm3 UCSC genome 2012 release and the position of the centromeres in the assembled ones. Again, we thank the reviewer for pointing out this information that was missing in our previous version of the manuscript.

d. I could not find anywhere how the average expression has been calculated to then obtain the list of genes with an expression higher than average.

We have reanalyzed our ChIPseq data in a different, more informative, way. Basically, we identify γ H2Av-enriched regions in the IPs samples (dH1-depleted and control) respect to the corresponding input samples and, additionally, called those regions showing at least a 3-fold higher coverage in dH1-depleted cells respect to control undepleted cells. In this way, we focused on the regions that are actually changing upon dH1-depletion. In this regard, γ H2Av regions localized at promoters of active genes were not identified because they did not change significantly upon dH1 depletion. Therefore, these results are no longer described in this revised version of the manuscript. Reanalysis of the ChIPseq data is described in the **Methods** section (page 18, lines 7-14), the **Results** section (page 5, second paragraph), **Figures 2a** and **b**, and in **Table 1**. Processed files and methods information under NCBI GEO accession GSE76912 have been updated appropriately.

3) Figure 1: b and c. In general, in all the ChIP data displayed the scale of the y axis is missing. Although I am sure it is consistent within one figure, it would be very important in order to compare data across different figures.

In this revised version we have included the scale in the y axis in **Figure 2b** (previous Figure 1b). We thank the reviewer for pointing out that this information was missing.

Looking at b. and also at Extended Data Figure 3b, the impression is that there is a general increase in the background of γ H2Av, and it is not so different between non-repetitive (Ext. Data Fig. 3b) and repetitive (Fig. 1b) regions.

As discussed above (second part of point 2), we have reanalyzed our ChIPseq data to identify γ H2Av-enriched regions showing at least 3-fold higher coverage in dH1-depleted cells. In this way, we identified 64 γ H2Av-regions specifically enriched in dH1-depleted cells. All of them localized to different types of repetitive elements. These results are described in the **Results** section (page 5, second paragraph), in **Figures 2a** and **b**, and in **Table 1**. Furthermore, our ChIP-qPCR results showed strong γ H2Av increase at multiple repetitive elements while major effects were observed at several non-repetitive regions analyzed. These results are described in the **Results** section (page 5, second paragraph) and in **Figure 2c**.

d. This data are not convincing. HP1a has a very wide localization. As the number of γ H2Av foci is increased upon H1 knock-down, the probability of HP1a- γ H2Av co-localization by chance would be higher than in the control cells.

As shown in **Figure 2d** (previous Figure 1d), we have compared co-localization in depleted and undepleted cells containing 1, 2, 3 or >3 foci independently. Therefore, the observed increased γ H2Av/HP1a overlap was independent of the actual increase of the number of γ H2Av foci in dH1-depleted cells. Furthermore, co-localization was higher in cells with few γ H2Av foci than in cells with >3 foci, indicating that the increased γ H2Av/HP1a overlap was not due to the increased

number of γ H2Av foci detected in dH1-depleted cells. These results are described in the **Results** section (page 5, third paragraph) and in **Figure 2d**.

In addition, is HP1a covered nuclear area comparable H1-knocked-down cells and controls? In this figure HP1 looks more extended in the knocked-down cells.

We observed that the total HP1a area increased upon dH1 depletion, but this increase was very weak and only respect to untreated cells and not respect to siRNA^{lacZ} cells (see Figure 1 below). Therefore, it cannot account for the observed increased overlap with HP1a. Nevertheless, we have recalculated all overlaps normalizing for the total HP1a area. These results are described in the **Methods** section (page 16, last three lines of second paragraph) and in **Figs 2d, 4b and 6e**.

Figure 1. Total HP1a area determined as the percentage of DAPI area stained with α HP1a antibodies is presented for siRNA^{dH1}, siRNA^{lacZ} and untreated cells.

e and f. In both cases the resolution is too low to claim that γ H2Av is somewhere specific.

The reviewer is right, the resolution does not allow to unambiguously demonstrated that γ H2Av signals and crossovers occur at any specific chromosomal location. Therefore, we have toned down these claims. These results are described in the **Results** section (page 5, last paragraph, lines 6-11, and page 6, second paragraph), and in **Figures 2e and 3a**.

4) Figure 2: a. This figure should be removed. Since it is shown in Extended Data Figure 6 that H1 knock-down causes an accumulation of cells in S-phase and according to Figure 3a the R-loops are much higher in S-phase irrespective of H1 depletion, concluding from cycling cells that R-loops are higher upon H1 knock-down is wrong.

I am afraid we do not agree. As shown in **Figure 6d** (previous Extended Data Figure 6d), dH1 depletion did not induce an increase of S-phase cells. Instead, we observed a weak G1-arrest (**Figure 6d**) and an increased number of EdU-positive cells at G1-phase (**Figure 6c**), suggesting that dH1-depleted cells initiated DNA replication prematurely. Therefore, G1-cells accounted for at least half of the S9.6 reactivity detected in cycling cells. As a matter of fact, the actual reactivity detected in cycling cells fits well with the reactivities, and proportion of cells, detected at the various phases. Therefore, we would like to keep the results shown in **Figure 4a** as a general overview of the observed effects on R-loops accumulation. We must admit that this part of the manuscript was poorly explained. In this revision, we try to describe more accurately these results in the **Results** section (page 6, fourth paragraph; page 7, last paragraph and page 8, first paragraph).

b. Here there is somehow the same problem as in Figure 1d. In the statistics to calculate the significance was it taken into consideration the probability of random co-localization due to the extent of HP1a staining and the increase in the number of S9.6 signals? In any case, the same figure for untreated cells should be shown.

Please, see third part of answer to point 3. Images of control undepleted cells have been included in **Figure 4b** (previous Figure 2b).

5) Extended Data Figure 6: a. It seems that in the control cells EdU and γ H2Av also co-localize.

The reviewer is right, in control cells there was also a significant co-localization between EdU and γ H2Av. However, what is relevant for our study is that the increased γ H2Av observed upon dH1 depletion occurred preferentially at EdU-

positive regions since the fold increase of the number of γ H2Av foci was significantly higher at EdU-positive than at EdU-negative regions. In this revised version, we describe these data in the **Results** section (page 8, first lines of first paragraph) and in **Supplementary Figure 5**.

d. There is no information of the statistics and the number of samples used for this figure. The differences are so minimal that it is important to explain how the statistical significance was obtained.

These data was obtained from three samples and, in total, more than 900.000 cells were sorted for each condition. In this revised version, this information is included in the **Legend to Figure 6d** (previous Extended Data Figure 6d).

6) Figure 3: a. The increase of γ H2Av between G1 and S-phase seems to be very similar between the control and H1 knock-down. Again, since there is no information of the statistics, I assume that the 3 asterisks refer to the difference between control and knock-down.

The reviewer is right, γ H2Av also increased in control undepleted cells in S-phase. As a matter of fact, γ H2Av is known to occur during normal DNA replication. However, what is relevant for our study is that, in dH1-depleted cells, γ H2Av in S-phase cells was much stronger than in control undepleted cells and strongly decreased upon RNH1 overexpression, while in control undepleted cells did not. The asterisks refer to the difference between dH1-depleted siRNA^{dH1} and control undepleted siRNA^{lacZ}. In this revised version, these results are described in the **Results** section (page 7, last paragraph and page 8, first paragraph) and in **Figure 5a**.

d. Presenting only the overly complicates the visualization.

In this revised version we present all three channels for each condition in **Figure 6e** (previous Figure 3d).

In addition, it would be much more informative to show a classification of replication patterns in the control and the corresponding (or not-corresponding) patterns in the si-RNA treated cells.

I am afraid we do not understand this point. We show representative examples of the patterns observed at the three subphases for each condition.

7) Figure 4: e. In the figure provided the levels of γ H2Av between untreated cells and cells treated with the si-RNA against HP1a look the same.

Indeed, γ H2Av levels did not change upon HP1a depletion, nor S9.6 reactivity did. It is precisely on the basis of these, and other results described in **Figure 7**, that we propose that, although relieves silencing, HP1a depletion does not induce R-loops accumulation in heterochromatin, suggesting that the contribution of dH1 to R-loops metabolism in heterochromatin is somehow specific.

Final comment

Finally, the sentence stating, " Altogether, these results support a model by which dH1 depletion relieves transcriptional silencing of heterochromatin regions, favoring R-loop FORMATION" should be changed. There is no evidence for an effect on formation. May be resolution? However, the only fair conclusion is that H1 is involved in R-loop metabolism.

We must fully agree with the reviewer, our results do not allow to discriminate between an effect on R-loops formation or resolution. In this revised version, we discuss this in more detail in the **Discussion** section (page 12, second paragraph) and in **Supplementary Table 1**. We have also revised carefully the text avoiding to talk about R-loops formation. We thank the reviewer for pointing this out.

REVIEWER #2**Main comments**

1) One of the important first bit of data regards the identification of H1-specific H2Av loci using ChIP-seq. According to Extended Fig 3, the authors have identified on the order of 15 such loci genome-wide. This seems extremely low and puts the relevance of the findings in question.

In the light of the reviewer's comment, we have reanalyzed our ChIPseq data in a different, more informative, way. Basically, we identify γ H2Av-enriched regions in the IPs samples (dH1-depleted and control) respect to the corresponding input samples and, additionally, called those regions showing at least a 3-fold higher coverage in dH1-depleted cells respect to control undepleted cells. In this way, we identified a higher number (N= 64) of γ H2Av-regions specifically enriched in dH1-depleted cells. All of them localized to different types of repetitive elements. These results are described in the **Results** section (page 5, second paragraph), in **Figures 2a** and **b**, and in **Table 1**. Furthermore, our ChIP-qPCR results showed strong γ H2Av increase at multiple repetitive elements while no major effects were observed at several non-repetitive regions analyzed. These results are described in the **Results** section (page 5, second paragraph, lines 11-14) and in **Figure 2c**. Details about the reanalysis of the ChIPseq data are provided in the **Methods** section (page 18, lines 7-14). We thank the reviewer for this comment that prompted us to reanalyze our ChIPseq data.

The statement that H1-independent peaks tend to map to TSS puzzles me a little as H2Av tracks, at least in mammalian systems, are very large (>100 kb) and can't be compared to TSS.

γ H2Av regions localized at promoters of active genes were not identified using the analysis described above since they did not change significantly upon dH1 depletion. Therefore, these results are no longer described in this revised version of the manuscript.

It is also unclear how many independent biological replicates of the ChIP-seq were performed, including input?

Our ChIPseq data comes from a single replicate for each condition. It must be noticed, however, that we confirmed the results by ChIP-qPCR at multiple repetitive elements as well as non-repetitive regions.

How did authors deal with problems of background subtraction and normalization over the repetitive portion of the genome?

In order to account for the repetitive nature of some genomic features and/or their multiple copies sparsed across the genome (i.e. transposons, tandem repeats, rDNA, etc), we chose an alignment policy that essentially allows for as many multiple hits as possible (setting an upper limit at 10.000 hits in order to somehow limit output file size). Since this alignment policy is used in both IPs and input control samples, identification of genomic features showing an enrichment proportion between them is still possible (see <http://www.ncbi.nlm.nih.gov/pubmed/27345571>). In this revised version, we described these analyses in the **Methods** section (page 17, last paragraph and page 18, first paragraph)

2) Another key argument put forth by the authors is that H1 depletion leads to R-loop accumulation. This is perhaps not entirely surprising but given the limited number of tools available to study R-loops, the data is not always convincing. One question raised here regards the amount of R-loops induced upon H1 depletion. The authors use microscopy for this and report the data as S9.6/DAPI (%area). In Fig 2a this number reaches 20% in dH1-depleted cells with a nice and small error bar. In Ext Fig 5 however, this number is now 40% with a similarly small error bar. Why the discrepancy? Is this saying that the method is in fact not quantitative?

The method we used is fairly reproducible. The difference between data described in **Figure 4a** (previous Figure 2a) and in **Supplementary Figure 4** (previous Ext Fig 5b) likely resides in the different experimental conditions of the two experiments. For the experiment in **Figure 4a** cells were transfected with a plasmid expressing RNH1 or with the corresponding empty plasmid. Instead, for

the experiment in **Supplementary Figure 4** cells were not transfected. Transfection is known to affect cellular physiology and, thus, is likely to have an impact on the extent of R-loops accumulation. Notice that, in both cases, we used the appropriate controls to compare the effect of dH1 depletion.

Regarding DRIP qPCR data, I would much prefer if the authors presented the data as % input on the y-axis. This bad habit of normalizing the data to a control completely prevents readers from gauging the frequency at which these structures may form.

As the reviewer mentions, the way we present the DRIP-qPCR data is widely used in the field (for recent publications see Castellano-Pozo et al., (2012) EMBO Rep, 13, 923; Herrera-Moyano et al., (2014) Genes Dev, 28, 735). Normalizing for a control region allows comparison of multiple regions, something that is particularly important in our study.

3) The authors also seem bent on showing that "that S9.6 foci significantly colocalize with HP1a in dH1-depleted cells". Yet, the data actually shows that only a very small portion of S9.6 signal colocalizes with HP1a based on Figure 2b and 2c. In conditions of H1 depletion, this colocalization may increase a bit but it remains marginal. I urge the authors to tone down their interpretations to better reflect reality.

The reviewer is right, S9.6 signals do not exclusively localized at HP1a-enriched regions. However, what is relevant for our study is that the overlap of S9.6 and HP1a signals strongly increased in dH1-depleted cells, suggesting that R-loops accumulated in heterochromatin. Furthermore, experiments in polytene chromosomes confirmed high S9.6 reactivity at heterochromatin upon dH1 depletion, while no reactivity was detectable in the euchromatic chromosome arms. In this revised version, we have rewritten this part of the manuscript and described these results in more detail in the **Results** section (page 6, last four lines and page 7, first two lines), **Figures 4b, 4c** and **Supplementary Figure 3**.

4) Throughout much of Figures 2 and 3, the authors use *RNASEH1* expression to show that the R-loop signal is "real" because it is sensitive to RNase H activity. This is pretty much standard operating procedure in the field despite accumulating evidence that *RNASEH1* expression is very toxic to cells, and may cause both transcriptional responses and impair DNA replication. I would urge the authors to study possible effects of *RNASEH1* expression on DNA replication more carefully as it is possible that the effects they are documenting are caused indirectly via a reduction in DNA replication caused by *RNASEH1*.

Treatment with RNH1 is widely accepted in the field as a way to analyze R-loops. Several observations suggest that, in *Drosophila* cells, this treatment does not have an important effect on DNA replication and cell viability. On one hand, the FACS profiles of cells expressing RNH1 or not show no major differences (see **Figure 1** below). Secondly, in wild-type flies, RNH1 overexpression in the wing imaginal disc does not affect wing development (see **Figure 2** below). Finally, and most important, our fiber analyses show that, in control undepleted cells, RNH1 expression does not significantly affect replication-fork progression (**Figure 6b**).

Figure 1. FACS profiles of dH1-depleted siRNA^{dH1} cells (b) and control untreated cells (a) expressing RNH1 (bottom) or not (top).

Figure 2. Wing of a wild-type fly expressing RNH1 in the wing imaginal disc
Then of course, when authors analyzed various phases of the cell cycle, the S9.6 signal was variably sensitive to RNASEH1, with S and G2/M phases showing RNASEH1-insensitive signal. This is extremely confusing and hard to fit in with any mechanism.

Our observation was that S9.6 reactivity at S-phase was largely unaffected by *in vivo* RNH1 expression. We agree with the reviewer that this was a puzzling observation. To address this question we have carried out different experiments. In particular, we have treated cells with bacterial RNH *in vitro* after sorting and we found that, upon treatment, S9.6 reactivity of S-phase cells strongly decreased, suggesting that, at least in part, it is associated with R-loops formation. In this revised version, we described these new results in the **Results** section (page 8, first paragraph) and in **Figure 5c** and **Supplementary Figure 6**. The reason of the low effect on S-phase S9.6 reactivity of *in vivo* RNH1 expression remains to be determined. It is possible that RNH1 activity, or its expression, is downregulated during S-phase. In this regard, there are two main classes of RNases H (RNH1 and RNH2), which are both capable of processing R-loops. In addition, RNH2 can also process single ribonucleotides in DNA and RNA primers during DNA replication. It has been proposed that, while RNH1 is the major player in removing transcription-associated R-loops, RNH2 might interact with the replication machinery and be involved in transactions during replication/repair processes{Chon, 2013 #108}. Thus, it is possible that RNH2 largely takes over the function of RNH1 in removing R-loops during S-phase. Further work is required to clarify this question. However, we think that this is out of the scope of our manuscript.

The suggestion by the authors that S9.6 recognizes RNA primers is doubtful and the reference 18 provides no evidence whatsoever for that claim - it must be deleted. If the authors want to claim that S9.6 recognizes Okazaki primers, they should provide direct evidence. Of course S9.6 has a documented minor affinity for AU-rich dsRNA but it is a bit unclear why such reactivity should obscure the signal only in S and G2/M.

The proposal that S9.6 antibodies might recognize RNA primers during replication was simply a proposal based on previous reports showing that S9.6 antibodies also recognize other nucleic acids structures (Phillips, et al., (2013) J Mol Recognit, 26, 376; Zhang et al., (2015) BMC Res Notes, 8:127). In this regard, our results suggest that S-phase S9.6 reactivity is not solely due to R-loops since it was not fully abolished by *in vitro* treatment with bacterial RNH and control undepleted cells showed higher reactivity in S-phase than in G1-phase. We must admit that this part of the manuscript was poorly explained. In this revised version, we tried to describe these results more accurately in the **Results** section (page 8, first paragraph) and in **Figure 5c** and **Supplementary Figure 6**. Further work is also required to fully clarify this question but, again, we think that this is out of the scope of our manuscript.

In the previous version of the manuscript the reference cited to support that S9.6 antibodies recognize various nucleic acids structures was an unfortunate mistake. We apologize for it. The correct references are cited in this revised version (Phillips, et al., (2013) J Mol Recognit, 26, 376; Zhang et al., (2015) BMC Res Notes, 8:127).

Other points

1) Often through the manuscript it is quite unclear how many independent experiments were performed or how many cells were analyzed to reach the conclusions reached by the authors. Similarly, some key technical details are sometimes missing. For ex. In Fig 1a, how many cells were analyzed? I would expect $n > 50$ to reach any kind of statistical confidence. In Extended Figure 2, how many times was this repeated? The absence of

error bars leads me to think this was done only once. In Fig 2a and 2b, how many cells were analyzed?

In this revised version, full details are provided in the **Figure Legends**, indicating in each case the number of independent experiments carried out, the sizes of the samples analyzed (always >50) and the statistical tests used. For instance in **Figure 1c** (previous Figure 1a) >100 cells were analyzed for each condition. Similarly in **Figure 1d** (previous Extended Figure 2), three independent experiments were performed and error bars, which were missing, are now shown, and in **Figures 4a** and **4b** (previous Figures 2a and 2b) >50 cells were analyzed for each condition.

2) In Fig 1d, the supposed co-localization between H2Av and HP1a is far from clear on the image provided.

I am afraid we do not agree. The image in **Figure 2d** (previous Figure 1d) for dH1-depleted cells shows three γ H2Av foci, 2 of which co-localize with HP1a. Instead for the controls the images show 1 or 2 foci that do not co-localize with HP1a.

3) In Fig 1e, the "intense α H2Ac reactivity observed at pericentric and telomeric heterochromatin in dH1-depleted chromosomes" seems at odd with the data in Fig 1d. Is that a property of metaphase chromosomes?

Our observation was that, while $\alpha\gamma$ H2Av reactivity was unusual in metaphase chromosomes from control undepleted cells, it was much more frequent in dH1-depleted metaphase chromosomes and, even in some cases, very intense $\alpha\gamma$ H2Av signals were observed. We must admit that this part of the manuscript was poorly explained. In this revised version, we tried to describe these results more accurately in the **Results** section (page 5, last paragraph) and in **Figure 2e** (previous Figure 1e) and **Supplementary Figure 2**.

4) In Fig 1f, the statement that increased SCE occur "preferentially at centromeric and telomeric regions" isn't supported. Provide data or delete.

The resolution of **Figure 3a** (previous Figure 1f) does not allow to unambiguously demonstrated that crossovers occur at any specific chromosomal location. Therefore, we have toned down this claim. These results are described in the **Results** section (page 6, second paragraph).

5) The statement that "55% of γ H2Av foci detected in dH1-depleted cells colocalize with S9.6 foci" isn't supported. Provide data or delete.

In this revised version these data are presented in **Figure 4a** (bottom graph on the right).

6) Fig 4 reports interesting genetic data using the wing model but it feels a little disconnect from the rest of the study. Likewise the observation of JNK mediated apoptosis is interesting but should also be studied in S2 cells.

I am afraid we do not agree. We think that the genetic data presented in **Figure 8** (previous figure 4) are important to analyze whether R-loops accumulation is on the basis of the strong apoptosis induced by dH1 depletion and to provide a molecular understanding of the essential role that dH1 plays in cell and organismal viability. We must admit, however, that this part of the manuscript was poorly explained. In this revised version, we tried to describe these results more accurately in the **Results** section (page 10, last paragraph and page 11, first three lines) and in **Figure 8** (previous Figures 4a and 4b).

7) In their model, the authors assume that firing origins in heterochromatin are more likely to break if they encounter R-loops. However, their data also show that heterochromatic regions shift to an earlier replication time in H1-depleted cells. This suggests that R-loops, directly or not, may favor origin firing. As the authors cite, there is some

interesting evidence that R-loops might be able to prime DNA synthesis. This should be added.

We fully agree with the reviewer and, in this revised version, we discuss this possibility in more detail in the **Discussion** section (page 11, last paragraph and page 12, first two lines).

Minor points

1) DNA breaks are referred to as DBs or DSBs, please be consistent

DBs corresponds to DNA breaks, which include both single-stranded DNA breaks (SSB) and double-stranded DNA breaks (DSB). We talk about DSB when we explicitly mean double-stranded breaks. This is a widely accepted nomenclature.

2) In Figure 2d, isn't the GATE LTR element upregulated in siRNA LacZ conditions

Figure 4d (previous Figure 2d) shows DRIP-qPCR analyses to analyze R-loops accumulation at several repetitive DNA elements, not gene expression.

REVIEWER #3

1) The Immunofluorescence images are in the most of the cases not very convincing. For example in Figure 1D at the siRNA^{dH1} condition there is no apparent increase in the overlap of γ -H2AX and the HP1 domain and the overlap might be due to the obvious increase of the HP1 domain possibly by decondensation. The same stands for figure 2B.

We observed that the total HP1a area increased upon dH1 depletion, but this increase was very weak and only respect to untreated cells and not respect to siRNA^{lacZ} cells (see Figure 1 below). Therefore, it cannot account for the observed increased overlap with HP1a. Nevertheless, we have recalculated all overlaps normalizing for the total HP1a area. These results are described in the **Methods** section (page 16, last three lines of second paragraph) and in **Figs 2d, 4b and 6e**.

Figure 1. Total HP1a area determined as the percentage of DAPI area stained with α HP1a antibodies is presented for siRNA^{dH1}, siRNA^{lacZ} and untreated cells.

Moreover images of 2A, 3A, 4D are of very low magnification and higher magnifications should be provided.

We think that it is important to show sections including several cells, which limits the magnification of the images considering the final size of the Figures. Fortunately, the current methodologies allow on line visualization of images at higher magnifications.

2) *The sister chromatid exchange depicted have to be tested in the presence of RNAase H to conclude that this phenotype is really the consequence of increased R loop formation in the absence of dH1.*

We have performed this experiment and found that in fact the frequency of SCE in dH1-depleted cells strongly decrease upon RNH1 expression. In this revised version, these data are described in the **Results** section (page 6, last two lines in fourth paragraph) and **Figure 3a**.

3) *Minor spelling mistakes*

We have carefully revised the manuscript looking for spelling mistakes.

Reviewers' Comments:

Reviewer #1 (Remarks to the Author)

Although the manuscript has improved and the authors have gone a good length to satisfy the comments, I still have some reservations, especially concerning the effects of H1 depletion in S-phase.

1. One major point is represented by the claim that there is EdU incorporation in G1 (first sentence page 9). It is not clear how this has been determined. It is clear that a claim of this sort requires identifying G1 in a manner that is independent of DNA content, for example staining for cyclin D or using MCM patterns or something similar. Given the data provided the claim is unsupported or very badly explained.
2. I still find highly unconvincing the biological significance of the G1 accumulation shown in Figure 6d.
3. I find the claim that there is anticipated replication of HP1 positive chromatin unconvincing. In the example provided in Figure 6e I cannot see a clear replicating HP1 positive region in early S-phase, while it is very clear in late.

The interpretation of the effects of histone H1 depletion on S-phase progression is overstated. For the manuscript to be deemed suitable for publication, this part needs to be profoundly revised, either in the text or adding supporting, independent experiments. I can agree with the model that formation (in G1?) of R-loop leads to checkpoint activation, that there are shorter tracts indicating slower fork progression (which could be a consequence of checkpoint activation) but I do not agree with the claims on replication timing and anticipated origin firing.

Reviewer #2 (Remarks to the Author)

In this revised manuscript the authors have addressed some of my issues but other remain and some have been amplified, raising significant concerns about the manuscript.

In the early part of the manuscript related to the mapping of gH2Av in dH1-depleted cells, a significant issue now stems from the acknowledgement that the gH2Av mapping by ChIP-seq has been derived from a single experiment (rebuttal letter). This is not acceptable, especially given that the read mapping strategy to repetitive elements is likely to generate a lot of noise. A second biological replicate is absolutely required to support this important piece of data.

The authors next argue that the increased gH2Av occurs over repetitive heterochromatin regions based on overlap with HP1 as measured from microscopy data. As mentioned in the previous round of review, the trouble with this experiment is that the HP1 signal is very broad and the resolution of the assay extremely low. Therefore the overlap between HP1 and gH2Av is necessarily high, raising questions as to the true meaning of this overlap. Authors next try to argue that this increased gH2Av occurs over regions marked by RNA:DNA hybrids using S9.6 signal, again using immunofluorescence experiments (Figure 4a). They claim that the overlap between both signals reaches near 60%. However, when one looks closely at the overlay image in dH1-depleted cells in Figure 4a, it is clear that the highest intensity signal for both gH2Av and S9.6 barely overlap and in fact appear to be exclusive of each other. Thus, while there seems to be an increase in both S9.6 and gH2Av, the relationship between both signals is in question. Further compounding the problem, the authors try to argue that the S9.6 signal also overlaps with HP1 in Figure 4b. However, even if the overlap increases somewhat from near zero in control cells, the larger conclusion of this experiment is that the large majority of the increased S9.6 signal (80%) in dH1 depleted cells does NOT overlap with HP1 and therefore heterochromatin. This suggests that the S9.6 signal response is not limited to heterochromatin and is in fact much broader. I now believe that the only way to truly measure the extent of the R-loop response in H1-depleted cells

and its precise relationship to HP1 signal and gH2Av signal is to perform global R-loop mapping using DRIP-seq. Without this, I worry that the statement "these results suggest that DNA damage and genomic instability induced by dH1 depletion is associated with R-loops accumulation in heterochromatin" is at best an over-simplification, at worst erroneous.

On the issue of measuring R-loops, the authors chose to ignore my simple request to present the DRIP-qPCR data as %input instead of normalizing to a negative control region. I must insist that they do so. As I mentioned in my previous review, normalizing to a negative control prevents readers from knowing the raw frequency at which authors detected R-loops. Are we talking about 0.1% or 10%? This is very informative as to how well the authors performed the experiment and the potential biological significance of the signal. Finally, showing the raw data would allow us to know that the relative increase they present is not due to a decrease of R-loops in the control region but rather to an increase in the test locus. At present, this can't be known. This is essential given that the effects here are small. The raw data might also explain why the ratios shown here with RNase H treatment go down to very low levels (0.1) which otherwise doesn't make sense. This data must be added at least to the supplementary material.

The section on the replication/cell cycle effects of dH1 depletion is probably the most interesting but suffers from being incomplete. I think the authors are sitting on a first-rate story but at that point their understanding of what's happening is too preliminary. For instance, there is an intrinsic contradiction in the observation that S phase cells S9.6 reactivity is not reduced by in vivo RNase H1 expression but can undergo "strong reduction" upon bacterial RNase H treatment in vitro. This, to me, indicates that a significant fraction of the S9.6 signal in S phase might be protected from RNase H1 action in vivo, which is potentially very interesting but completely not understood. A second contradictory finding in the manuscript is that RNase H1 expression in vivo apparently corrects the replication asymmetry problem in dH1-depleted cells, arguing that RNase H1 does affect something. Either RNase H1 has an effect independent of R-loops (because S9.6 signal is not affected), which would put in question the use of RNase H1 as a tool, or it is through R-loops but then the lack of S9.6 signal change goes unexplained. This is murky. Likewise, the proposal that "R-loops induced by dH1 depletion accumulate at G1-phase, cause DNA damage during S-phase and affect the pattern of DNA replication" is interesting but completely lacks in mechanistic understanding. Are the authors suggesting that R-loops induced in G1 last until S phase? This seems unlikely given recent data (PMID: 27373332) that R-loops in mammalian systems show a half life of 10 minutes. The dynamic of R-loops in the *Drosophila* system should be tested.

In the discussion, the authors fail to explain why transcriptional upregulation causes R-loops in dH1-depleted cells but not in HP1-depleted cells, even though this is one of the most striking results of the study. In their model, the authors also completely drop any mention of aberrant R-loops priming DNA replication, shifting heterochromatic regions towards an earlier replication patterns and potentially triggering many of the phenotypes that they observed, including replicating stress from uncoordinated replication origin firing. Again, this is one of the most striking (if still poorly understood) aspects of the study.

On a more minor point, the manuscript (and letter) is riddled with grammatical and spelling mistakes, making it hard to follow in some places.

Altogether, this manuscript presents some interesting and novel observations on the role of the linker histone H1 in regulating R-loop metabolism and its impact on replication timing and genome stability. At this stage however, the manuscript still shows significant deficiencies in some key sections having to do with gH2Av and R-loop mapping, which results in a lack of mechanistic understanding of the phenotypes triggered by dH1 depletion.

REVIEWER #1

General comments

The reviewer "**still have some reservations concerning the effects of H1 depletion in S-phase**" and, in the final remark, the reviewer states that "**this part needs to be profoundly revised, either in the text or adding supporting, independent experiments**".

We must agree with the reviewer that additional experiments are needed to reach a better understanding of the contribution of dH1 to DNA replication. Genome-wide mapping of the effects of dH1 depletion on the temporal and spatial pattern of DNA replication appears essential. Similarly, analyzing dH1 dynamics during DNA replication, its release before and deposition after fork passage is also of great importance. However, we think that these studies, which would constitute an extremely interesting story by themselves, are currently out of the scope of our paper that focuses on the role of dH1 in preventing R-loops-induced DNA damage and genomic instability in heterochromatin. Thus, following the reviewer's suggestions, we have profoundly revised this part in both the Results (page 8, last paragraph and page 9) and the Discussion sections (page 13, last paragraph and page 14, first paragraph), and Figures have been modified accordingly (Figures 4c, 4d and 6).

In the final remark, the reviewer "**agrees with the model that formation of R-loop leads to checkpoint activation**" and "**that there are shorter tracts indicating slower fork progression**", and proposes that the later "**could be a consequence of checkpoint activation**".

We agree with the reviewer that this is the most likely possibility. Results from our genome-wide DRIPseq analysis confirmed that dH1 depletion induces R-loops accumulation in heterochromatin and, moreover, indicate that dH1 depletion does not induce R-loops accumulation globally. Thus, it is unlikely that the global slow down of DNA replication observed in dH1-depleted cells is due to direct collisions with R-loops related lesions. On the other hand, it is well established that checkpoint activation, or uncoordinated DNA replication, slow down DNA replication globally. In this regard, the rescue of the replication defects observed upon RNH1 expression are likely indirect; the removal of R-loops prevents the generation of R-loops-induced DSB during DNA replication in heterochromatin and, thus, the checkpoint is not activated and replication progresses normally. This possibility is discussed in page 14, first paragraph. DRIPseq analysis is described in Fig. 5 and Supplementary Fig. 5.

Also in the final remark, the reviewer "**does not agree with the claims on replication timing and anticipated origin firing**".

*Our evidence for early heterochromatin replication is based on IF experiments suggesting increased EdU incorporation in heterochromatin in early S-phase, as shown by the higher EdU/HP1a overlap observed in dH1-depleted cells in comparison to control undepleted cells. Concomitantly, we observed a reduced Edu/HP1a overlap in late S-phase in dH1-depleted cells. In this regard, the reviewer "**didn't see in the images provided a clear replicating HP1***

positive region in early S-phase” (see specific comment 3). The reviewer might be right of this and we provide additional images in Figure 6b to further illustrate our observations. However, although the actual EdU/HP1a overlap of individual cells is somehow variable, our results are based on the quantitative analysis of a large number of cells (described in Materials and Methods) and are statistically significant (p -value < 0.005). Concerning the possibility that R-loops induced in heterochromatin by dH1-depletion might initiate DNA replication, we must agree with the reviewer that it is just a (appealing) speculation. In this revised version we specifically state in the Discussion section the speculative nature of this possibility (page 14, first paragraph).

Specific comments

1. One major point is represented by the claim that there is EdU incorporation in G1 (first sentence page 9). It is not clear how this has been determined. It is clear that a claim of this sort requires identifying G1 in a manner that is independent of DNA content, for example staining for cyclin D or using MCM patterns or something similar. Given the data provided the claim is unsupported or very badly explained.

and

2. I still find highly unconvincing the biological significance of the G1 accumulation shown in Figure 6d.

The reviewer is right. Following the reviewer’s suggestion we have profoundly revised this part of the manuscript and these results have been deleted since they do not directly address our main observation that dH1-depletion induces R-loops accumulation in heterochromatin that cause replicative stress and genomic instability.

3. I find the claim that there is anticipated replication of HP1 positive chromatin unconvincing. In the example provided in Figure 6e I cannot see a clear replicating HP1 positive region in early S-phase, while it is very clear in late.

Please, see answer to the last general comment above.

REVIEWER #2

Main comment

In the final remark, the reviewer states: “**At this stage, the manuscript still shows significant deficiencies in some key sections having to do with gH2Av and R-loop mapping**”.

*To address these concerns, we have performed additional γ H2Av ChIPseq and DRIPseq analyses. For γ H2Av ChIPseq, we performed two new independent replicates for control and dH1-depleted conditions. In these new experiments, we detected a high number of γ H2Av-enriched regions in both control and dH1-depleted cells. A large proportion of the identified regions were common to both conditions, mapping to promoters of actively transcribed genes, which is in agreement with the known role that γ H2Av plays in transcription regulation in *Drosophila*, where H2Av is the single H2A variant that acts as both mammalian H2A.X and H2A.Z. To identify regions where γ H2Av content specifically increased in dH1-depleted cells, we used DiffBind/edgeR. This analysis resulted in the identification of 166 regions, of which ~70% corresponded to repetitive DNA elements, both transposable elements (TE) and simple repeat satellite DNAs. Permutation analysis showed that this enrichment was highly statistically significant (p -value < 0.0002). These results, which are in good agreement with our previous ChIPseq experiment, confirmed that γ H2Av specifically induced by dH1-depletion preferentially accumulates in heterochromatin. Notice that, to avoid technical differences associated with deep sequencing, the results from our previous ChIPseq experiments have not been included in the analysis presented in this revised version, which is based on the two new ChIPseq replicates that were processed and subjected to deep-sequencing in parallel. These results are presented in Fig. 2, Supplementary Fig. 2 and Table 1, and are discussed in the Results section (page 5, last paragraph and page 6, first paragraph).*

For DRIPseq, we performed two independent replicates for control and dH1-depleted cells that were treated or not with bacterial RNH before IP with S9.6 antibodies. In these experiments, we detected a high number of enriched genomic regions in both control and dH1-depleted cells that were sensitive to RNH. These regions distributed along the entire genome, which is in good agreement with results in mammalian cells showing extensive R-loop formation across the genome. Similar to γ H2Av, a large proportion of the R-loops enriched regions detected were common to both control and dH1-depleted cells. In this case, we also used DiffBind/edgeR to determine regions where R-loops abundance specifically increased in dH1-depleted cells, resulting in the identification of 189 regions that, like for γ H2Av, were highly enriched in repetitive DNA elements (~95% of them corresponded to TE and satellite DNAs) (p -value < 0.0002).

Specific comments

1. In the early part of the manuscript related to the mapping of gH2Av in dH1-depleted cells, a significant issue now stems from the acknowledgement that the gH2Av mapping by ChIP-seq has been derived from a single experiment (rebuttal letter). This is not acceptable,

especially given that the read mapping strategy to repetitive elements is likely to generate a lot of noise. A second biological replicate is absolutely required to support this important piece of data.

We have performed two new independent replicates for control and dH1-depleted conditions, which confirmed that γ H2Av specifically induced by dH1-depletion preferentially accumulates in heterochromatin (please, see first paragraph of the answer to the main comment above). γ H2Av ChIP-qPCR data fully agree with these results.

2. The authors next argue that the increased gH2Av occurs over repetitive heterochromatin regions based on overlap with HP1 as measured from microscopy data. As mentioned in the previous round of review, the trouble with this experiment is that the HP1 signal is very broad and the resolution of the assay extremely low. Therefore the overlap between HP1 and gH2Av is necessarily high, raising questions as to the true meaning of this overlap.

Our conclusion that γ H2Av specifically induced by dH1-depletion preferentially accumulates in heterochromatin is mainly based on the γ H2Av ChIPseq results. From this point of view, IF experiments are confirmatory. These results show a significantly higher γ H2Av/HP1a overlap in dH1-depleted cells than in control undepleted cells. Although HP1a area is broad, it is so in both dH1-depleted cells and control undepleted cells and, thus, it should not seriously affect our observation since what it is really relevant for our study is that the γ H2Av/HP1a overlap significantly increased upon dH1-depletion, not the actual γ H2Av/HP1a overlap observed. Notice that, to account for a possible increased HP1a area upon dH1-depletion, the extent of overlap was normalized for the total HP1a area. In addition, the highest γ H2Av/HP1a overlap increase was observed in cells containing one γ H2Av foci and, thus, was not reflecting random colocalization due to the increased number of γ H2Av foci detected in dH1-depleted cells. In this revised version, we tried to explain better these results, clearly stating the confirmatory nature of the IF experiments. These results are now presented in Supplementary Fig. 3, being discussed in the Results section (page 6, last paragraph).

Authors next try to argue that this increased gH2Av occurs over regions marked by RNA:DNA hybrids using S9.6 signal, again using immunofluorescence experiments (Figure 4a). They claim that the overlap between both signals reaches near 60%. However, when one looks closely at the overlay image in dH1-depleted cells in Figure 4a, it is clear that the highest intensity signal for both gH2Av and S9.6 barely overlap and in fact appear to be exclusive of each other. Thus, while there seems to be an increase in both S9.6 and gH2Av, the relationship between both signals is in question.

In the previous version, these data was presented as the proportion of γ H2Av foci overlapping with S9.6 signal, which might be a bit confusing since it did not distinguish between foci highly overlapping with S9.6 and those showing weak overlap. In this revised version, we determined γ H2Av/S9.6 overlap in the same way as for the rest of IF experiments by measuring the proportion of total γ H2Av

area overlapping with S9.6 signal (γ H2A/S9.6 area overlap), which is certainly more accurate. In this way, the observed overlap is slightly less than 40%. Moreover, we must again emphasize that the relevance of these results reside in the fact that the overlap was significantly higher in dH1-depleted cells than in control cells (p -value < 0.005), not in the actual overlap observed. It is also important that γ H2Av signal is abolished by RNH1 expression, suggesting that it depends on R-loops. In this revised version, we tried to explain better these results. These data are presented in Fig. 4b, bottom panel.

Further compounding the problem, the authors try to argue that the S9.6 signal also overlaps with HP1 in Figure 4b. However, even if the overlap increases somewhat from near zero in control cells, the larger conclusion of this experiment is that the large majority of the increased S9.6 signal (80%) in dH1 depleted cells does NOT overlap with HP1 and therefore heterochromatin. This suggests that the S9.6 signal response is not limited to heterochromatin and is in fact much broader. I now believe that the only way to truly measure the extent of the R-loop response in H1-depleted cells and its precise relationship to HP1 signal and γ H2Av signal is to perform global R-loop mapping using DRIP-seq.

To address these concerns, we have performed DRIPseq analyses with two biological replicates. These studies confirmed that R-loops specifically induced by dH1-depletion preferentially accumulate in heterochromatin (please, see second paragraph of the answer to the main comment above). These studies showed extensive R-loops formation across the entire genome in both control and dH1-depleted cells. As a matter of fact, most R-loops enriched regions were detected in both conditions. However, R-loops enriched regions detected specifically in dH1-depleted cells map to heterochromatic elements. These observations not only confirmed the accumulation of R-loops in heterochromatin upon dH1 depletion, but they also account for the extent of S9.6/HP1a overlap observed by IF. Since R-loops distribute along the entire genome, not only in heterochromatin, S9.6 reactivity is not constrained to HP1a regions. As a matter of fact, the actual S9.6/HP1a overlap is pretty low in control cells. Upon dH1 depletion, R-loops specifically accumulate in heterochromatin, while non-heterochromatic R-loops are largely unaffected, which results in a significantly higher S9.6/HP1a overlap in dH1-depleted cells than in control cells. DRIPseq analyses are presented in Fig. 5, Supplementary Fig. 5 and Table 1, and are discussed in the Results section (page 7, last paragraph and page 8, first paragraph). IF experiments are presented in Supplementary Fig. 6, where additional images have been incorporated to further illustrate our observations, and are discussed in the Results section (page 8, last sentence of first paragraph).

Regarding DRIPseq analyses, it does not escape our observation that these data also contained general information about naturally occurring R-loops and their potential biological significance. However, we believe that a detailed analysis and description of this information is out of the scope of this manuscript, which focuses on the specific effects of dH1 depletion on R-loops formation. Genome-wide analysis of where R-loops occur naturally, and what

role(s) they play, would be a different (exciting) story that has been the subject of several recent studies in mammalian and yeast cells.

3. On the issue of measuring R-loops, the authors chose to ignore my simple request to present the DRIP-qPCR data as %input instead of normalizing to a negative control region. I must insist that they do so. As I mentioned in my previous review, normalizing to a negative control prevents readers from knowing the raw frequency at which authors detected R-loops. Are we talking about 0.1% or 10%? This is very informative as to how well the authors performed the experiment and the potential biological significance of the signal. Finally, showing the raw data would allow us to know that the relative increase they present is not due to a decrease of R-loops in the control region but rather to an increase in the test locus. At present, this can't be known. This is essential given that the effects here are small. The raw data might also explain why the ratios shown here with RNase H treatment go down to very low levels (0.1) which otherwise doesn't make sense. This data must be added at least to the supplementary material.

We agree with the reviewer that normalizing for a negative control region might mask a change associates with a decrease in the negative control region, instead of an increase in the analyzed regions. Thus, in this revised version, we present the DRIP-qPCR data without normalizing for any of the two negative control regions that we analyzed (MTHL12 and Tubulin), showing no big effects at the negative control regions and, thus, similar results. The data presented as DRIP efficiency (%INPUT) is included below (Figure 1). As it can be observed, DRIP efficiencies of the samples not treated with RNH were usually higher than 0.1. Of course, after treatment with RNH, DRIP efficiencies dropped to very low levels. Still, we think that it is much more informative (and easy to follow for the reader) to present data as fold-change over untreated cells, which is what actually matters, than as DRIP efficiency (once confirmed that they were reasonably OK). Note also that the statistical significance of the differences were determined respect to the control siRNA^{lacZ} cells, to account for effects related with activation of the RNAi pathway. We think that adding the data presented as DRIP efficiency to the supplementary information would be odd. These results are presented in Fig. 5c and are discussed in the Results section (page 8, last part of first paragraph)

Figure 1. DRIP-qPCR data presented as DRIP efficiency (%INPUT)

4. The section on the replication/cell cycle effects of dH1 depletion is probably the most interesting but suffers from being incomplete. I think the authors are sitting on a first-rate story but at that point their

understanding of what's happening is too preliminary. For instance, there is an intrinsic contradiction in the observation that S phase cells S9.6 reactivity is not reduced by in vivo RNase H1 expression but can undergo "strong reduction" upon bacterial RNase H treatment in vitro. This, to me, indicates that a significant fraction of the S9.6 signal in S phase might be protected from RNase H1 action in vivo, which is potentially very interesting but completely not understood.

We agree with the reviewer that this was a puzzling observation. Why is S9.6 reactivity in S-phase sensitive to treatment with bacterial RNH in vitro, but insensitive when human RNH1 is transiently expressed in vivo? We are afraid we cannot provide a final answer. Sensitivity to RNH in vitro, strongly suggest that, at least in part, it is due to R-loops and, consequently, that human RNH1 does not efficiently target R-loops in S-phase Drosophila cells. Perhaps, RNH1 activity is tightly regulated in S-phase to control degradation of DNA:RNA hybrids formed during DNA replication and human RNH1 misses this regulation in Drosophila cells. Actually, RNH1 overexpression in wild-type flies does not cause major developmental defects, which suggest inefficient targeting of DNA:RNA hybrids in S-phase since, if efficient, RNH1 overexpression must compromise their metabolism during DNA replication and, thus, affect development. On the other hand, part of the S9.6 reactivity observed in S-phase is likely due to dsRNAs since it is sensitive to RNase A. These results are presented in Figure 4c and Supplementary Figs. 7 and 9, and are discussed in the Results (page 9, first paragraph) and the Discussion sections (page 13, first paragraph). While it is evident that further work is required to clarify this aspect, we think that what is really relevant for our work is that S9.6 reactivity is detected in G1-phase, before γ H2Av increases in S-phase, and that, at G1, S9.6 reactivity is abolished by RNH1 expression.

A second contradictory finding in the manuscript is that RNase H1 expression in vivo apparently corrects the replication asymmetry problem in dh1-depleted cells, arguing that RNase H1 does affect something. Either RNase H1 has an effect independent of R-loops (because S9.6 signal is not affected), which would put in question the use of RNase H1 as a tool, or it is through R-loops but then the lack of S9.6 signal change goes unexplained. This is murky.

R-loops induced by dh1 depletion slow down DNA replication globally. Most likely, this effect is an indirect consequence of checkpoint activation due to the abnormal R-loops accumulation and DNA damage that dh1 depletion induces in heterochromatin since no global R-loops accumulation, or γ H2AV increase, was detected. From this point of view, the rescue of the replication defects observed upon RNH1 expression are likely indirect too; the removal of R-loops in G1 prevents the generation of R-loops-induced DSB during DNA replication (please, see below) and, thus, the checkpoint is not activated and replication progresses normally. This possibility is discussed in page 14, first paragraph.

Likewise, the proposal that "R-loops induced by dh1 depletion accumulate at G1-phase, cause DNA damage during S-phase and affect the pattern of DNA replication" is interesting but completely lacks in

mechanistic understanding. Are the authors suggesting that R-loops induced in G1 last until S phase? This seems unlikely given recent data (PAID: 27373332) that R-loops in mammalian systems show a half life of 10 minutes. The dynamic of R-loops in the Drosophila system should be tested.

Several mechanisms could account for the replicative stress caused by R-loops formed during G1-phase. On one hand, targeting of the ssDNA in the R-loop by DNA damaging agents and/or processing of the DNA:RNA hybrid by NER could cause SSBs that would convert into DBS during DNA replication. On the other hand, if dH1 is involved in recruitment of R-loops destabilizing factors (please, see below), its depletion could significantly extend the half-life of R-loops to reach S-phase. These possibilities are presented in Fig. 9 and are discussed in page 12, second paragraph.

5. In the discussion, the authors fail to explain why transcriptional up regulation causes R-loops in dH1-depleted cells but not in HP1-depleted cells, even though this is one of the most striking results of the study.

In this revised version, we have performed additional ChIP-qPCR experiments that provide further support for the specific contribution of dH1 to the regulation of R-loops metabolism in heterochromatin. These results show that HP1a depletion, which relieves heterochromatin silencing but does not induce R-loops accumulation in heterochromatin, does not decrease dH1 occupancy at heterochromatic elements. In HP1a-depleted cells, R-loops accumulate in heterochromatin only when dH1 is also co-depleted. We also present results showing that dH1 depletion reduces HP1a occupancy in heterochromatin, which is in agreement with results reported by others indicating a role of dH1 in the recruitment of Su(vary)3-9. Altogether these results suggest that relief of heterochromatin silencing per se is not sufficient to induce R-loops accumulation and that the effect of dH1 is specific. In this regard, dH1 has been shown to interact with several factors that could be involved in preventing and/or resolving R-loops. We hypothesize that dH1 might be involved in the recruitment to heterochromatin of such factors, some of which are known to localize to heterochromatin and be required for silencing. Further work is required to support this hypothesis. We agree with the reviewer that deciphering the actual mechanism(s) by which dH1 prevent R-loops accumulation would be an exciting follow-up of our current work. These results are presented in Fig. 7, Supplementary Fig. 8 and Supplementary Table 1, and they are discussed in the Results (page 10) and Discussion sections (page 12, last two sentences of the first paragraph.

In their model, the authors also completely drop any mention of aberrant R-loops priming DNA replication, shifting heterochromatic regions towards an earlier replication patterns and potentially triggering many of the phenotypes that they observed, including replicating stress from uncoordinated replication origin firing. Again, this is one of the most striking (if still poorly understood) aspects of the study.

The possibility that R-loops induced in heterochromatin by dH1-depletion might induce unscheduled initiation of DNA replication in heterochromatin, though

speculative, is very appealing. In this revised version, we clearly state this possibility in the Discussion section (first paragraph in page 14).

6. On a more minor point, the manuscript (and letter) is riddled with grammatical and spelling mistakes, making it hard to follow in some places.

We have carefully revised the manuscript trying to fix grammatical and spelling errors. We apologize for any mistakes that may still be present.

Reviewers' Comments:

Reviewer #1:

Remarks to the Author:

Bayona-Feliu and co-authors have considerably improved the data presented. There are still modifications to be made, listed below. However, upon correction, the manuscript can be published.

Corrections needed:

1. One general criticism is that the figures are not cited in order. Going back and forth is very inconvenient. Example Fig.4 and 5.
2. Page 3, in three different sentences, metazoan is used incorrectly. It is an adjective, for example metazoan organisms. The correct word is metazoa. Similarly, in the sentence "...eukaryotes contain linker histone H1..." sounds like prokaryotes have only core histones. Eliminate eukaryotes. In general, there are English inaccuracies throughout: what is high organismal lethality? Either is lethal or not. May be penetrance?
3. Page 4: "R-loops are three—stranded structures formed when an RNA transcript invades the DNA duplex...". It is wrong. The "R-loops are three—stranded structures formed when the newly transcribed RNA forms a hybrid molecule with ...".
4. Supplemental Fig. 2d: the accumulation of phosphorylated H2Av on promoters of actively transcribed genes can also be interpreted as a consequence of transcription (see sentence in page 5). It is also confusing that while in Supplemental Fig. 2d the genes that are not enriched for phosphorylated H2Av have a low expression, the ones in Supplemental Fig. 2f have instead a higher expression.
5. As the residue of H2Av phosphorylated upon DNA damage and concomitantly with transcription is the same, an accumulation of phosphorylated H2Av upon H1 depletion could be attributed to increased expression. Maybe it should be shown that this accumulation depends on the active DNA damage checkpoint, while the transcriptional-dependent phosphorylation does not.
6. Fig. 2a: please in supplemental provide the bar graph as % of input, not only has % of base pairs.
7. Supplemental Figure 3a shows the overlap between S9.6 and HP1a signals, not phosphorylated H2Av.
8. Supplemental Fig. 3x: the scale to say that the signal is proximal to centromere is so big that you could also interpret the data as the phosphorylated H2Av being excluded from centromeres!
9. Fig. 3a: no claim can be made regarding crossover at centromeres. There is no way to identify properly the centromere in these images.
10. Fig. 4b, bottom graph: there is no explanation on how the overlap between phosphorylated H2Av and S9.6 signal was calculated. The overlap looks very variable (see also Supplemental Fig. 4). How can the error bars be so small?
11. Fig. 4d: how does it come that the increase of phosphorylated H2Av in G1 is significant in c but not in d?
12. Fig. 5 d is c in reality. How many times was the experiment repeated. From what I understand is one experiment with technical duplicates? This is not enough.
13. Supplemental Fig. 6: "S9.6 reactivity significantly overlapped with Hp1a..". The data are presented wrongly. In untreated cells, there is no S9.6 signal therefore there can be nothing but an increase in the overlap with HP1a upon signal increasing. It looks like there is about 1/5 of the S9.6 signal overlapping with HP1a signal. Is this in agreement with the DRIP data? Please show an inset in Supplemental Fig. 6b.
14. Any comment about the very high background for H1 ChIP in Fig. 7e?
15. Fig. 7d: The reduction of HP1 binding in cells treated with H1siRNA is shown as highly significant but it is not evident at all in any of the IF.
16. Supplemental Fig. 8b: an IF on untreated polytene chromosomes needs to be shown.
17. One problem with the model is that in all the suggested cases phosphorylated H2Av would require passage through S-phase, as the SSB gets converted into DSB. Instead, although there is

an increase in S, the signal is already visible in G1.

Reviewer #2:

Remarks to the Author:

The authors have significantly improved the manuscript by performing additional ChIP-seq and DRIP-seq experiments. Overall, the study is likely to generate a fair amount of discussion but I now feel comfortable that this discussion will occur with the backdrop of significant experimental data for others to re-analyze.

In that vein, I would encourage the authors to take advantage of the transparent peer-review process and make the reviews and rebuttal public. I think many in the field would appreciate seeing our exchanges, including the authors' thoughtful replies.

In the Discussion, I find it very odd that the authors have not included the findings of Susan Gasser's group (PMID: 27668659) on repeat-mediated instability in the absence of H3K9 methylation. While the nature of centromeres in *C. elegans* is quite distinct, this seems very relevant to the discussion on the respective roles of H3K9 methylation, HP1 and histone H1 in R-loop suppression over repeated regions of the genome. I would encourage the authors to include this in a brief revision.

Minor comments:

- Please change DRIPseq and ChIPseq to DRIP-seq and ChIP-seq as is convention.
- The manuscript, while improved, will still need significant proofreading for english especially in the new parts.

REVIEWER #1

1. One general criticism is that the figures are not cited in order. Going back and forth is very inconvenient. Example Fig.4 and 5.

Following the reviewer's suggestion, we have rewritten this part of the manuscript so that, in this revised version, results presented in Fig. 4 are described in full before going to Fig. 5. (Page 7, third paragraph and page 8, second paragraph).

2. Page 3, in three different sentences, metazoan is used incorrectly. It is an adjective, for example metazoan organisms. The correct word is metazoa. Similarly, in the sentence "...eukaryotes contain linker histone H1..." sounds like prokaryotes have only core histones. Eliminate eukaryotes. In general, there are English inaccuracies throughout: what is high organismal lethality? Either is lethal or not. May be penetrance?
and

3. Page 4: "R-loops are three—stranded structures formed when an RNA transcript invades the DNA duplex...". It is wrong. The "R-loops are three—stranded structures formed when the newly transcribed RNA forms a hybrid molecule with ...".

We have fixed these inaccuracies and revised the rest of the text.

4. Supplemental Fig. 2d: the accumulation of phosphorylated H2Av on promoters of actively transcribed genes can also be interpreted as a consequence of transcription (see sentence in page 5). It is also confusing that while in Supplemental Fig. 2d the genes that are not enriched for phosphorylated H2Av have a low expression, the ones in Supplemental Fig. 2f have instead a higher expression.

Data presented in Supplemental Figure 2 was a mix. On one hand, panels a, b, c and d showed the analysis of γ H2Av enriched regions in control and dH1-depleted cells, while panels e and f showed the properties of the genomics regions where γ H2Av was specifically decreasing in dH1-depleted cells with respect to control cells. In this way, panel d showed that γ H2Av target genes detected in both control and dH1-depleted had a higher average expression in S2 cells than non-target genes. On the other hand, panels e and f showed that the regions where γ H2Av specifically decreased in dH1-depleted cells corresponded to promoters (panel e) of genes that show high expression in S2 cells (panel f). We agree with the reviewer that this way of presenting the data was confusing. In this revised version we have split these data in two different Supplemental Figures. Supplemental Figure 2 presents the data corresponding to panels a, b, c and d, while Supplemental Figure 3 presents the data of panels e and f.

5. As the residue of H2Av phosphorylated upon DNA damage and concomitantly with transcription is the same, an accumulation of phosphorylated H2Av upon H1 depletion could be attributed to increased expression. Maybe it should be shown that this accumulation depends on the active DNA damage checkpoint, while the transcriptional-dependent phosphorylation does not.

Data presented in Figure 8c show that γ H2Av induced by dH1 depletion depends on dATR/mei-41 since it is abolished by dATR co-depletion. Additional data in Figure 8 also show that co-depletion of other DDR components rescue the apoptotic phenotype induced by dH1 depletion. These results indicate that, indeed, γ H2Av induced by dH1 depletion depends on check-point activation. In good agreement, dH1 depletion induces a high incidence of DNA breaks (Figure 1c). Furthermore, γ H2Av remains low in G1-phase, when most genes are transcribed, while it strongly increases during S-phase, when only a subset of genes are transcribed (Figure 4c), suggesting that γ H2Av induced by dH1-depletion is not associated with a global effect on transcription that, on the other hand, is known to be rather weak (references 11 and 15).

6. Fig. 2a: please in supplemental provide the bar graph as % of input, not only has % of base pairs.

We understand the reviewer is asking for how many (%) of the regions where γ H2Av increases upon dH1 depletion contain repetitive DNA sequences, which is ~70% (N= 166). This is explicitly stated in the text (page 5, beginning of the last sentence). We do not think necessary to make a graph out of these data.

7. Supplemental Figure 3a shows the overlap between S9.6 and HP1a signals, not phosphorylated H2Av.

This has been corrected (Supplemental Figure 4a). We thank the reviewer for pointing out this error.

8. Supplemental Fig. 3c: the scale to say that the signal is proximal to centromere is so big that you could also interpret the data as the phosphorylated H2Av being excluded from centromeres!

Following the reviewer's comment, we have rewritten this sentence (page 6, last sentence of the second paragraph).

9. Fig. 3a: no claim can be made regarding crossover at centromeres. There is no way to identify properly the centromere in these images.

In many cases, it looks like the crossover occurred at centromeric regions. However, we agree that the resolution is too low to tell the precise sites of the crossovers. This claim has been deleted (page 6, last paragraph).

10. Fig. 4b, bottom graph: there is no explanation on how the overlap between phosphorylated H2Av and S9.6 signal was calculated. The overlap looks very variable (see also Supplemental Fig. 4). How can the error bars be so small?

We thank the reviewer for pointing this out since the way overlaps were determined was not properly described. In this revised version, this is described in the Methods section (page 17, last two sentences of the first paragraph). Like the rest of overlaps, the γ H2Av/S9.6 overlap was determined based on maximal RGB projections of the corresponding 0.5 μ m thick confocal stacks and was expressed as the proportion of γ H2Av area overlapping with S9.6. Data on Figure 4b, bottom graph, correspond to the analysis of >50 cells for each condition. The small error bars indicate that the overlap is not so variable when a sufficiently large number of cells are analyzed.

11. Fig. 4d: how does it come that the increase of phosphorylated H2Av in G1 is significant in c but not in d?

Panels c and d present data obtained by two different methodologies, IF (panel c) and WB (panel d). In both cases, a weak γ H2Av increase was detected in G1. However, while this increase was statistically significant in the IF experiments, it was not in the WB experiments. This is likely because of the larger number of cases analyzed by IF (>50 cells for each condition) in comparison to WB (two independent experiments).

12. Fig. 5 d is c in reality. How many times was the experiment repeated. From what I understand is one experiment with technical duplicates? This is not enough.

Data presented in Figure 5c (thanks for pointing out this error) correspond to triplicates of two independent biological replicates, as indicated in the Methods section (page 21, third paragraph) and in the Legend to Figure 5.

13. Supplemental Fig. 6: “S9.6 reactivity significantly overlapped with Hp1a..”. The data are presented wrongly. In untreated cells, there is no S9.6 signal therefore there can be nothing but an increase in the overlap with HP1a upon signal increasing. It looks like there is about 1/5 of the S9.6 signal overlapping with HP1a signal. Is this in agreement with the DRIP data? Please show an inset in Supplemental Fig. 6b.

We agree with the reviewer that, at this point, we cannot discard the possibility that the higher S9.6/HP1a overlap is simply the consequence of the increased S9.6 reactivity observed in dH1-depleted cells. Interestingly, when our DRIP-seq data was crossed with HP1a ChIP-seq data obtained in larvae (modENCODE 4936), we observed that 20% of the R-loops enriched regions in S2 cells are targeted by HP1a (see Figure 1 below). In this revised version, we have removed panel a in Supplemental Fig. 7 (Supplemental Fig. 6 in the earlier version), while we keep panel b showing RNH-sensitive S9.6 reactivity at the heterochromatic chromocenter in dH1-depleted polytene chromosomes.

Figure 1. Euler diagram showing the overlap between S9.6 peaks detected in dH1-depleted cells and HP1a peaks (modENCODE 4936)

14. Any comment about the very high background for H1 ChIP in Fig. 7e?

We understand that the reviewer refers to the 50% reduction in signal observed in dH1-depleted cells. Whether this reflects background or residual dH1 is uncertain since the extent of depletion is not exactly the same in all experiments and it never reaches 100% (see Figure 2 below). In any case, what

is relevant for our work is that HP1a depletion did not affect dH1 occupancy in experiments where dH1 depletion showed a significant effect.

15. Fig. 7d: The reduction of HP1 binding in cells treated with H1siRNA is shown as highly significant but it is not evident at all in any of the IF.

Though significant, the reduction in HP1a occupancy at heterochromatic elements detected by ChIP is weak (~25% in most of the regions analyzed). This reduction might not be detectable by IF since the total HP1a content determined by WB (see Figure 2 below) as well as the total HP1a area determined by IF (see Figure 3 below) are not substantially altered upon dH1 depletion. In this regard, it has been shown that dH1 depletion in flies results in the mobilization of HP1a from a single chromocenter to multiple HP1a enriched regions (reference 10) (see also Supplemental Figure 8). Altogether these observations suggest that, although dH1 depletion reduces HP1a occupancy in heterochromatin, the global HP1a immunostaining might not be equally affected.

Figure 2. WB analysis of increasing amount of extracts prepared from control undepleted cells and dH1-depleted cells with α dH1, α HP1a and α H4 antibodies.

Figure 3. Total HP1a area determined as the percentage of DAPI area stained with α HP1a antibodies is presented for siRNA^{dH1}, siRNA^{lacZ} and untreated cells.

16. Supplemental Fig. 8b: an IF on untreated polytene chromosomes needs to be shown.

Images of control undepleted polytene chromosome are included in Supplemental Fig. 9 (Supplemental Fig. 8 in the earlier version of the manuscript).

17. One problem with the model is that in all the suggested cases phosphorylated H2Av would require passage through S-phase, as the SSB gets converted into DSB. Instead, although there is an increase in S, the signal is already visible in G1.

It is well established that DSB can also occur during G1-phase, being generally repaired by NHEJ. It is possible that, though with low frequency, R-loops already generate DSB in G1 since DNA insults can occur in both the unpaired strand and, for instance, at the junctions between the DNA:RNA hybrid and the DNA duplex (these regions are likely distorted and, thus, more sensitive to damaging agents). However, what is relevant for our work is that the bulk of the DNA damage induced by R-loops occurs in S-phase.

REVIEWER #2

The reviewer encourages us **"to take advantage of the transparent peer-review process and make the reviews and rebuttal public. I think many in the field would appreciate seeing our exchanges, including the authors' thoughtful replies"**

Definitely, we will opt for the transparent-peer review scheme

The reviewer finds it very odd that, in the Discussion, we **"have not included the findings of Susan Gasser's group (PMID: 27668659) on repeat-mediated instability in the absence of H3K9 methylation. While the nature of centromeres in *C. elegans* is quite distinct, this seems very relevant to the discussion on the respective roles of H3K9 methylation, HP1 and histone H1 in R-loop suppression over repeated regions of the genome. I would encourage the authors to include this in a brief revision"**

In this revised version we include a sentence on this work (page 11, third sentence of the second paragraph)

Minor comments:

-Please change DRIPseq and CHIPseq to DRIP-seq and ChIP-seq as is convention.

We have done these changes

-The manuscript, while improved, will still need significant proofreading for english especially in the new parts.

We have revised one more time the whole manuscript